# From Noise to Control: Parameterized Diffusion Policies

**Renhao Zhang** [1]  **Haotian Fu** [2]  **Mingxi Jia** [2]  **George Konidaris** [2]  **Yilun Du** [3]  **Bruno Castro da Silva** [1]

## Abstract

We propose Parameterized Diffusion Policy (PDP), a framework for learning diffusion policies conditioned on low-dimensional, continuous parameters embedded in a learned behavior manifold. By constructing this manifold so that distances between latent representations reflect the semantic similarity between physical trajectories, we transform diffusion from a mechanism for stochastic diversity into a precise and optimizable tool for behavior steering. Our approach enables smooth interpolation between known strategies and efficient adaptation to novel constraints without updating policy weights. We demonstrate that PDP significantly improves adaptation performance on complex multimodal benchmarks in both simulated and real-robot experiments compared to standard diffusion policies, particularly in scenarios requiring the synthesis of novel behaviors.

## 1. Introduction

Real-world robot tasks are often *underspecified*: a given goal in a particular environment may be achieved by many distinct action sequences (Ivanovic et al., 2020; Lynch et al., 2020). A soccer-playing robot, for example, may score via multiple feasible shot trajectories that all satisfy "ball enters goal" but differ in how they avoid defenders and satisfy constraints (Figure 1). This one-to-many structure is pervasive in manipulation and navigation (Mandlekar et al., 2021; Florence et al., 2022; Chen et al., 2023). Consequently, expert datasets provided for robot policy learning are naturally *multimodal*, and an ideal policy is expected to both (i) incorporate such multimodal behaviors and (ii) find the *right* behavior to execute when constraints change (da Silva et al., 2012; Dalal et al., 2021).

[1]University of Massachusetts [2]Brown University [3]Harvard University. Correspondence to: Renhao Zhang <renhaozhang@cs.umass.edu>.

*Proceedings of the 43rd International Conference on Machine Learning*, Seoul, South Korea. PMLR 306, 2026. Copyright 2026 by the author(s).

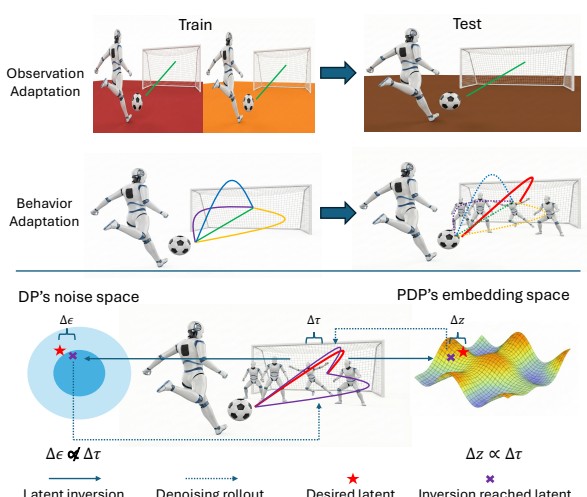

*Figure 1.* **Observation-side shift vs. constraint-induced behavior shift, and why PDP enables stable behavior steering.** Top: Standard evaluations vary observations while the intended behavior remains unchanged. Middle: We study constraint-induced behavior shifts, where environmental changes invalidate many of the trajectory modes in the training dataset, and success requires selecting or discovering a different trajectory mode. Bottom: In standard DP, steering via the sampling noise is ill-conditioned: small perturbations can lead to large, unpredictable trajectory changes. PDP, by contrast, exposes a learned behavior latent $z$ whose geometry is aligned with trajectory similarity, so small changes in $z$ induce smooth changes in behavior, enabling interpolation and adaptation.

Diffusion policies have recently become a strong default for behavior cloning because they can model multimodal and high-dimensional action sequences and produce diverse rollouts via stochastic denoising (Chi et al., 2023). However, diversity alone is not the same as *controllability*. Most standard evaluations predominantly focus on *observation-side* distribution shift (Figure 1, top) (Zhu et al., 2023; Wolf et al., 2025), i.e., ensuring that changes in appearance or camera conditions do not affect the physically intended behavior. In contrast, many deployment failures arise from *constraint-induced behavior shifts* (Figure 1, middle): a new obstacle, contact constraint, or feasibility condition can invalidate previously demonstrated strategies, forcing the robot to select or discover a novel behavior mode beyond those covered in the training dataset (Konidaris & Barto, 2009; Pan et al., 2025a; Wagenmaker et al., 2025).

This paper targets the following setting: a diffusion policy is pretrained offline on multimodal demonstrations. At deploy-

ment time, the task remains the same, but *constraints shift* so that only one or a few of the behavior modes covered in the training dataset remain viable. In harder cases, none of the observed modes may satisfy the new constraints, requiring the agent to interpolate a new behavior mode. Crucially, we wish adaptation to be *fast* and *data-light*, potentially guided by only a single new demonstration. This motivates two requirements that current diffusion-policy control interfaces do not satisfy:

**(1) Fast behavior steering via parameter updates.** When constraints unseen in training invalidate existing strategies, we would like to steer the policy at test time by adapting only a *small set of parameters*, rather than retraining the full policy. In other words, adaptation should require only solving a compact optimization problem, or taking a few gradient steps, to produce a feasible behavior that reliably satisfies the new constraints.

**(2) A smooth, geometry-aligned behavior manifold for generalization.** Beyond selecting among memorized modes, the policy should generalize *within* the training distribution by interpolating between behaviors to produce plausible trajectories that were not explicitly demonstrated. This calls for a behavior space whose geometry is aligned with trajectory similarity: moving a small distance in this space should induce a correspondingly small, predictable change in the executed trajectory. Such a space would support both mode interpolation (to improve coverage) and rapid adaptation to new constraints (to preserve feasibility).

We propose **Parameterized Diffusion Policies (PDP)**: diffusion policies conditioned on a *continuous* behavior parameter $z$ learned from demonstrations. PDP learns a trajectory encoder that maps each demonstration to a latent code and optimizes the resulting representation so that it is *geometry-aligned*: distances in $z$ reflect semantic similarity between physical trajectories. Conditioning the diffusion denoiser on $z$ converts diffusion from "noise-driven diversity" into an explicit and optimizable control mechanism: $z$ becomes a low-dimensional handle for (i) selecting a desired mode, (ii) smoothly interpolating between behaviors to produce unobserved but in-distribution trajectories, and (iii) rapidly adapting to constraint shifts by optimizing $z$ at test time, without updating policy weights. We evaluate PDP in a benchmark suite designed specifically to isolate behavior-side adaptation under constraint shifts with multimodal training data. Across multiple simulation tasks and a real robot task, we find that PDP enables more reliable steering and adaptation than other diffusion policy and behavior cloning baselines.

## 2. Related Work

**Diffusion Models and Latent Structure.** Diffusion models have become a dominant class of generative models because they can represent complex, multimodal distribu-

tions, but their latent spaces are often poorly structured and their control signals can be difficult to optimize. This limits their ability to smoothly interpolate and control outputs in both vision and sequential decision-making settings (Sohl-Dickstein et al., 2015; Song & Ermon, 2019; Ho et al., 2020; Dhariwal & Nichol, 2021; Park et al., 2023; Pan et al., 2025b; Hahm et al., 2024; Moser et al., 2025; Karras et al., 2022; Nichol & Dhariwal, 2021; Song et al., 2021; Li et al., 2025a;b). Recent analyses have highlighted geometric distortions in diffusion latent spaces and proposed disentangled or isometric latent learning to improve interpolation and editability (Park et al., 2023; Pan et al., 2025b; Hahm et al., 2024). These limitations are even more consequential for control and decision tasks, where precise behavior steering is essential for safety and feasibility. Consequently, recent surveys have framed such limitations as key opportunities for diffusion in robotics (Zhu et al., 2023; Xu et al., 2025; Wolf et al., 2025; Makarova et al., 2025), while other works have explored RL-guided training to align generative behaviors with objectives beyond simple likelihood (Black et al., 2024; Miao et al., 2024; Wang et al., 2025).

**Diffusion Policies and Behavior Steering.** Building on generative diffusion models, diffusion policies have emerged as a powerful paradigm for robot imitation (Chi et al., 2023; Wang et al., 2023; Li et al., 2024b; Jackson et al., 2024; Janner et al., 2022; Høeg et al., 2025). While expressive, these policies often require test-time steering to adapt to novel constraints without retraining (Du et al., 2023b; Yu et al., 2025). Current strategies typically steer a frozen policy by injecting external guidance, such as value-function gradients, human feedback, or specialized geometric priors (Ding et al., 2024; Li et al., 2025c; Shan et al., 2025; Du et al., 2023a; Du & Mordatch, 2019; Zhu et al., 2024). Other methods optimize directly in the high-dimensional noise space to refine behaviors (Wagenmaker et al., 2025; Li et al., 2024a; Kang et al., 2023; Ding et al., 2025). However, this optimization is often ill-conditioned due to the lack of geometric regularization in the latent noise (Park et al., 2023; Pan et al., 2025b; Hahm et al., 2024). To improve controllability, a parallel line of work learns explicit representations, often through discrete bottlenecks such as skill tokens or vector-quantized codes (Chen et al., 2023; Lee et al., 2024; Wu et al., 2025; Qiao et al., 2025; Ma et al., 2025). While these representations facilitate mode selection, they naturally restrict control to a finite behavior set.

**Parameterized Skills and Actions.** Similar high-level ideas about parameterized actions and skills have also been studied in reinforcement learning as mechanisms for structured abstraction and task transfer (Masson et al., 2016; Hausknecht & Stone, 2016; Dalal et al., 2021; Zhang et al., 2024; Gupta et al., 2025). By learning explicit hierarchical representations, these methods improve sample efficiency and enable the composition of complex behaviors in long-

horizon tasks (Fu et al., 2023; Zheng et al., 2024; Konidaris & Barto, 2009; Queisser & Steil, 2018; Sutton et al., 1999; Frans et al., 2018; Vezhnevets et al., 2017). PDP extends these classical concepts to the modern generative paradigm by learning a continuous behavior-semantic manifold that parameterizes the diffusion denoising process.

## 3. Background

**Diffusion models.** Let $x_0 \in \mathbb{R}^D$ denote a data vector (e.g., an action chunk with dimension $D$). Denoising diffusion probabilistic models (DDPMs) define a forward noising process (Ho et al., 2020):

$$q(x_k \mid x_{k-1}) = \mathcal{N}(\sqrt{\alpha_k}\, x_{k-1},\, (1 - \alpha_k)I), \qquad (1)$$

where $k = 1, \ldots, K$ indexes the diffusion timestep, and $x_k$ denotes the state of the sample after applying $k$ steps of the forward diffusion process to $x_0$. Let $\bar{\alpha}_k \triangleq \prod_{s=1}^{k} \alpha_s$. Then, the marginal has the following closed form:

$$x_k = \sqrt{\bar{\alpha}_k}\, x_0 + \sqrt{1 - \bar{\alpha}_k}\, \epsilon, \qquad \epsilon \sim \mathcal{N}(0, I). \qquad (2)$$

A conditional diffusion model learns a noise predictor $\epsilon_\theta(x_k, k, c)$, where $c$ denotes an arbitrary conditioning variable, using the standard noise-prediction objective:

$$\mathcal{L}_{\text{DDPM}}(\theta) = \mathbb{E}_{x_0, k, \epsilon}\left[\|\epsilon - \epsilon_\theta(x_k, k, c)\|_2^2\right]. \qquad (3)$$

**Diffusion policies for action chunks.** A robot demonstration trajectory is $\tau = \{(o_t, a_t)\}_{t=0}^{T-1}$, where $o_t$ is an observation and $a_t \in \mathbb{R}^{d_a}$ is a continuous action. Diffusion policies model *action chunks* of horizon $H$ (Chi et al., 2023), denoted by $A_t = \{a_t, a_{t+1}, \ldots, a_{t+H-1}\}$, conditioned on a context feature $c_t$, such as an encoding of recent observations. A diffusion policy $\pi_\theta(A_t \mid c_t)$ instantiates (2)–(3) with $x_0 \equiv A_t$, producing chunks by iterative denoising and executing them in a receding-horizon fashion.

**Test-time inversion via latent/noise optimization.** Given a trained generative model $G_\theta(\cdot)$, including diffusion models, a standard adaptation mechanism is to optimize an input variable $u$ (e.g., a noise seed, a conditioning vector, or an internal latent) while keeping $\theta$ fixed. In particular, given a target behavior $\tilde{\tau}$, define the discrepancy loss as $\mathcal{L}_{\text{fit}}(u; \tilde{\tau})$ and solve $u^\star = \arg\min_u \mathcal{L}_{\text{fit}}(u; \tilde{\tau}) + \lambda \mathcal{R}(u)$, where $\mathcal{R}$ enforces a prior or feasibility bias on $u$ (e.g., $\|u\|_2^2$).

## 4. Method

We introduce **Parameterized Diffusion Policy (PDP)**, which extends the diffusion policy framework by incorporating a continuous behavior latent variable $z \in \mathbb{R}^{d_z}$ to explicitly model the multimodality of the training data. PDP factorizes control into two components: (i) a geometry-aligned behavior manifold that maps high-dimensional demonstrations to a latent space where Euclidean distances reflect

trajectory similarity, and (ii) a parameterized diffusion denoiser that generates action sequences conditioned on both environmental context and a target latent variable $z$. Unlike standard diffusion policies, which rely on stochastic noise for diversity, PDP exposes $z$ as a low-dimensional control handle. At test time, this interface enables precise mode selection, smooth behavior interpolation, and rapid adaptation to novel constraints by optimizing only the latent $z$, while keeping the underlying policy parameters fixed (Fig. 2).

### 4.1. Learning a Geometry-Aligned Behavior Latent

PDP centers on learning a latent manifold that explicitly models the multimodality of the training data, such that the distance between two latent codes $z_i$ and $z_j$ reflects the semantic similarity of their corresponding physical behaviors. Rather than structuring this space directly on raw observations, we extract a behavior trace from each demonstration $\tau$: a state-based feature sequence that captures the essential geometric and functional structure of the task execution (e.g., positional states). Mapping such traces to continuous latent codes $z$ transforms the high-dimensional variability of demonstrations into a structured coordinate chart suitable for precise behavior steering and interpolation.

**Behavior Embedding Architecture.** To map the complex variability of demonstrations into a continuous latent space, we learn a trajectory encoder $E_\phi$. This encoder maps each full trajectory $\tau$ to a Gaussian posterior over latent codes:

$$q_\phi(z \mid \tau) = \mathcal{N}(\mu_\phi(\tau),\, \text{diag}(\sigma_\phi^2(\tau))). \qquad (4)$$

For downstream policy conditioning, we use the deterministic mean representation $\bar{z}(\tau) \triangleq \mu_\phi(\tau)$. Although the encoder processes the complete demonstration, a symmetric decoder $D_\psi$ is employed during training to reconstruct the original trajectory $\tau$ from $z$, ensuring that the latent space preserves all necessary behavioral information.

**Joint Embedding Objective.** The encoder-decoder $(\phi, \psi)$ is optimized via a multi-objective loss that balances information density, distributional regularity, and metric alignment:

$$\mathcal{L}_{\text{embed}} = \mathcal{L}_{\text{rec}} + \beta_{\text{KL}}\mathcal{L}_{\text{KL}} + \beta_{\text{geo}}\mathcal{L}_{\text{geo}}. \qquad (5)$$

The first two terms correspond to the standard VAE evidence lower bound (ELBO). The **reconstruction loss**, $\mathcal{L}_{\text{rec}} = \mathbb{E}_{z \sim q_\phi(\cdot \mid \tau)}\left[\|\tau - D_\psi(z)\|_2^2\right]$, trains $z$ to capture the salient features of the demonstration, while the **prior regularization**, $\mathcal{L}_{\text{KL}} = D_{\text{KL}}(q_\phi(z \mid \tau) \,\|\, \mathcal{N}(0, I))$, shapes the latent distribution toward a standard normal prior, facilitating test-time sampling and preventing the manifold from collapsing.

**Geometry Alignment via Soft-DTW.** Recall that standard VAEs do not guarantee that latent proximity corresponds to behavioral similarity. To make $z$ a *smooth* space for search and interpolation, we explicitly align the Euclidean geometry in the latent space with the physical geometry of the trajectories. In particular, for each demonstration

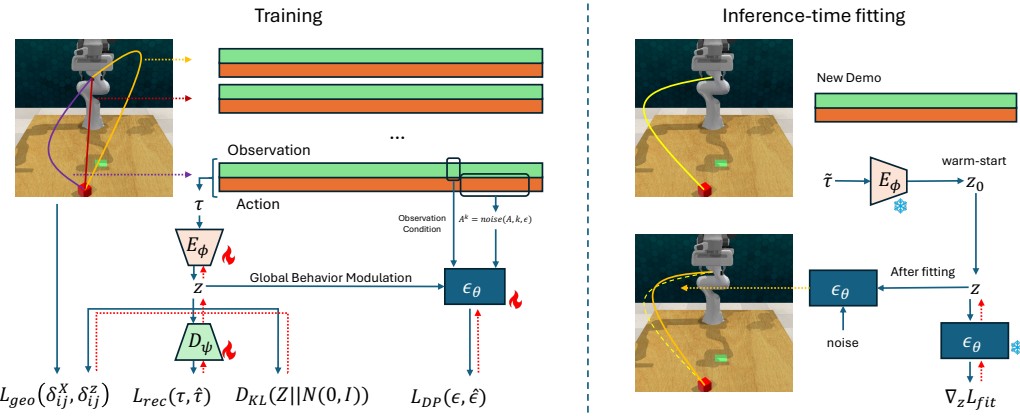

*Figure 2.* **PDP framework. Training (left):** The trajectory encoder $E_\phi$ embeds each demonstration $\tau$ into a behavior latent code $z$ sampled from a Gaussian posterior. The latent representation is optimized via a joint objective: a standard VAE loss (reconstruction $\mathcal{L}_{rec}$ and KL-divergence $D_{KL}$) preserves information and regularizes the latent distribution, while a geometry loss $\mathcal{L}_{geo}$ aligns latent distances $\delta_{ij}^z$ with physical trajectory similarities $\delta_{ij}^X$ computed via soft-DTW. In parallel, the denoiser $\epsilon_\theta$ is trained to predict the noise $\epsilon$ added to action chunks $A^k$, conditioning on environmental observations and $z$ through global modulation. **Inference-time fitting (right):** Given a new demonstration $\tilde{\tau}$, the frozen encoder $E_\phi$ provides a semantic warm-start $z_0$. This latent code is iteratively refined by minimizing the fitting loss $\mathcal{L}_{fit}$ via gradient descent ($\nabla_z \mathcal{L}_{fit}$) through the frozen denoiser. After fitting, the optimized $z$ is used to condition the denoiser $\epsilon_\theta$, which generates the full action sequence by iteratively denoising a Gaussian seed into a trajectory adapted to the new constraints.

$\tau$, we extract representative features $X(\tau)$, such as positional features in the proprioception state. We then use **Dynamic Time Warping (DTW)** on $X(\tau)$ to compute trajectory distances, since DTW provides a distance metric **invariant to temporal variations and execution speed**.

Unlike standard $L_2$ norms, which are sensitive to time shifts, DTW identifies an optimal alignment between two sequences, $A = (a_1, ..., a_n)$ and $B = (b_1, ..., b_m)$, by matching points such that endpoint alignment and monotonicity are preserved. Given a local cost matrix $D$, where $D_{i,j} = \Delta(a_i, b_j)$, the cumulative cost $r_{i,j}$ is computed via the recurrence $r_{i,j} = D_{i,j} + \min\{r_{i-1,j}, r_{i,j-1}, r_{i-1,j-1}\}$, where the final distance is $r_{n,m}$. Since the min operator is non-differentiable, we employ the soft-DTW relaxation. This replaces the hard minimum with a smoothed operator: $\text{softmin}_\gamma(x_1, ..., x_k) = -\gamma \log \sum_{i=1}^{k} \exp(-x_i/\gamma)$, allowing gradients to flow back to the encoder. A more comprehensive analysis is provided in Appendix A.1.

Next, for each pair of demonstrations $(\tau_i, \tau_j)$, we define the trajectory distance $\delta_{ij}^X$ and the latent distance $\delta_{ij}^z$ as follows:

$$\delta_{ij}^X \triangleq d_{\text{softDTW}}^\gamma \big(X(\tau_i), X(\tau_j)\big), \quad \delta_{ij}^z \triangleq \|\bar{z}(\tau_i) - \bar{z}(\tau_j)\|_2. \quad (6)$$

The **geometry loss** $\mathcal{L}_{geo}$ (Eq. (5)) enforces a linear correspondence between these distances:

$$\mathcal{L}_{geo} = \mathbb{E}_{(i,j)\sim\mathcal{P}} \big[ \big( \delta_{ij}^z - \kappa \delta_{ij}^X \big)^2 \big], \quad (7)$$

where $\kappa$ is a scaling constant and $\mathcal{P}$ is the distribution over sampled trajectory pairs. This alignment ensures the manifold is navigable, supporting smooth interpolation and predictable steering. Appendix A.4 analyzes how the geometry-

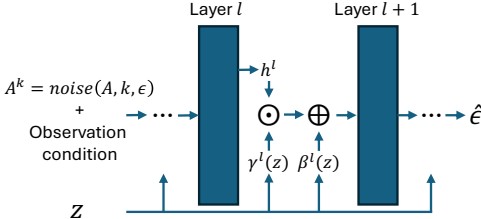

*Figure 3.* **Global Modulation for Denoiser.** The behavior latent code $z$ is transformed by MLPs into layer-specific parameters $\gamma^{(l)}$ and $\beta^{(l)}$. These are applied to feature maps $h^{(l)}$ via the affine transformation $h^{(l)} \leftarrow \gamma^{(l)}(z) \odot h^{(l)} + \beta^{(l)}(z)$.

aligned latent structure facilitates training and stabilizes low-dimensional adaptation.

### 4.2. Learning Parameterized Diffusion Policies

With a structured behavior manifold established, we now train a diffusion policy $\pi_\theta$ that generates action chunks $A_t$ conditioned on both the environmental context $c_t$ and the behavior latent code $z$. Unlike standard diffusion policies, which rely on stochastic noise as an implicit source of multimodality, PDP treats $z$ as an explicit control signal that determines the mode and style of the generated trajectory.

**Latent-Conditioned Denoising via Global Modulation.** As discussed in Section 3, we model the policy as an iterative denoising process. Each action chunk $A_t$ within a demonstration $\tau$ is associated with the behavior latent code $\bar{z}(\tau)$ produced by the frozen encoder $E_\phi$. We train a neural denoiser $\epsilon_\theta$ to predict the noise $\epsilon$ added to the action chunk

at diffusion step $k$:

$$\hat{\epsilon} = \epsilon_\theta(A_t^k, k, c_t, \bar{z}), \qquad \bar{z} \triangleq \text{sg}(\mu_\phi(\tau)). \quad (8)$$

This formulation incorporates two critical design choices for effective behavior steering: (1) **Decoupled Representation Learning:** the use of the stop-gradient operator $\text{sg}(\cdot)$ is essential to decouple the geometric shaping of the behavior manifold from denoiser training. By treating $\bar{z}$ as a fixed input, we prevent the diffusion loss (which is primarily concerned with mode fitting) from collapsing or distorting the distance-preserving properties enforced by $\mathcal{L}_{\text{geo}}$. (2) **Deep Behavior Integration:** Rather than treating $\bar{z}$ as a standard input feature, we implement the denoiser as a modulated neural field. The behavior latent code deeply reconfigures the network's internal score field via a modulation architecture inspired by FiLM (Perez et al., 2018). As illustrated in Figure 3, this mechanism applies affine transformations to intermediate feature tensors $h^{(\ell)}$ at multiple layers: $h^{(\ell)} \leftarrow \gamma^{(\ell)}(\bar{z}) \odot h^{(\ell)} + \beta^{(\ell)}(\bar{z})$, where $\gamma^{(\ell)}$ and $\beta^{(\ell)}$ are lightweight MLPs. This hierarchical modulation provides a robust handle for shifting the output distribution across distinct behavioral modes, ensuring that low-dimensional latent codes effectively steer high-dimensional action generation. Our ablation studies confirm that this deep integration is crucial, as naive concatenation is often ignored by the denoiser.

**Training and Inference.** The policy parameters $\theta$ are optimized by minimizing the latent-conditioned noise-prediction error:

$$\mathcal{L}_{\text{DP}}(\theta) = \mathbb{E}_{\tau, t, k, \epsilon}\left[\|\epsilon - \epsilon_\theta(A_t^k, k, c_t, \bar{z}(\tau))\|_2^2\right]. \quad (9)$$

We use a **joint training** scheme in which we alternate between (i) updating the embedding parameters $(\phi, \psi)$ using $\mathcal{L}_{\text{embed}}$ and (ii) updating the policy parameters $\theta$ using $\mathcal{L}_{\text{DP}}$. A detailed procedural breakdown of this scheme is provided in Appendix A.2.

### 4.3. Test-Time Control and Latent Adaptation

Once trained, PDP provides a continuous, optimizable interface for steering the policy without updating policy weights. While standard diffusion policies rely on stochastic sampling to discover behaviors, PDP treats the behavior manifold as a structured "search space". The high-level intuition is to replace unstructured noise sampling with gradient-based optimization in latent parameter space to identify the specific latent code $z$ that best satisfies the new environmental constraints encountered at deployment time. This change turns test-time adaptation into a more stable, well-conditioned refinement process.

**Latent Fitting with Encoder Warm-Start.** When environmental constraints shift or a specific target behavior $\tilde{\tau}$ is required, we infer the optimal latent code $z^*$ that aligns the (frozen) diffusion policy with the given requirements. Unlike prior inversion methods that optimize in high-dimensional noise spaces or unregularized latent spaces, we

use our geometry-aligned encoder to provide a *semantic warm-start*: $z_0 = \mu_\phi(X(\tilde{\tau}))$. This initialization places $z$ near the relevant region of the behavior manifold, improving optimization efficiency and stability. Starting from $z_0$, we refine the latent code by minimizing a diffusion-consistent fitting objective:

$$\mathcal{L}_{\text{fit}}(z; \tilde{\tau}) = \mathbb{E}_{t, k, \epsilon}\left[\|\epsilon - \epsilon_\theta(\tilde{A}_t^k, k, \tilde{c}_t, z)\|_2^2\right] + \lambda\|z\|_2^2, \quad (10)$$

where $\tilde{A}_t^k$ denotes the action chunk from the target demonstration noised according to the forward process. The first term in Eq. (10) identifies the latent code under which the frozen denoiser best matches the target actions, while the quadratic regularizer ensures $z$ remains within the support of the training distribution. We then solve for the optimal latent code via gradient descent:

$$z \leftarrow z - \eta \nabla_z \mathcal{L}_{\text{fit}}(z; \tilde{\tau}), \quad (11)$$

typically running $S$ steps before execution to obtain a precise control handle for the resulting rollout. The complete test-time latent adaptation procedure is formalized in Appendix A.3.

**Long-Horizon Adaptation via Segmented Latents.** For complex tasks consisting of distinct functional phases, a single global latent code may be insufficient to capture all required behavioral variation. PDP naturally extends to these settings by supporting piecewise-constant latent sequences. In particular, given a segmented demonstration $\tilde{\tau} = \{\tilde{\tau}^{(m)}\}_{m=1}^M$, we fit an independent latent code $z^{(m)}$ for each phase $m$ by minimizing the joint objective:

$$\min_{z^{(1)}, \ldots, z^{(M)}} \sum_{m=1}^M \mathcal{L}_{\text{fit}}(z^{(m)}; \tilde{\tau}^{(m)}). \quad (12)$$

This allows the policy to transition through a sequence of behavior-specific regions of the latent manifold, enabling the synthesis of complex, long-horizon maneuvers while maintaining the interpretability and steerability of the individual latent segments.

## 5. Experiments

Our experiments are designed to systematically evaluate the capacity of PDP to transform diffusion-based behavior cloning into a controllable and adaptable control framework. We evaluate four key properties: (i) **effective behavior steering**, demonstrating PDP's ability to reliably adapt by selecting or synthesizing feasible strategies under environmental constraint shifts; (ii) **original-task fidelity**, demonstrating that behavior latent codes are sufficiently expressive to reconstruct multimodal distributions present in the training data; (iii) **geometry-driven generalization**, analyzing how the structured latent space enables smooth behavior interpolation; and (iv) **architectural efficacy**, verifying through

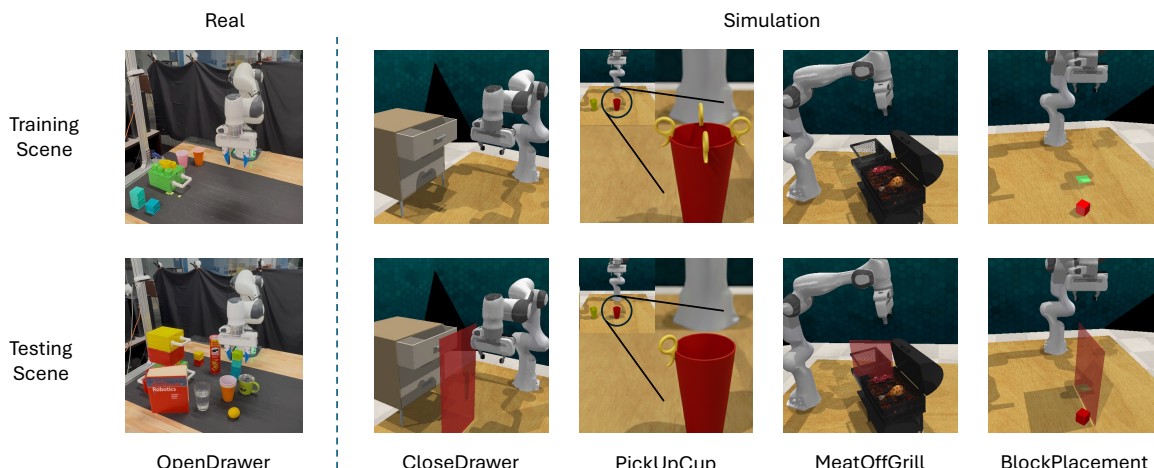

*Figure 4.* **Benchmark domains for evaluating controllable multimodal imitation. Top row**: Training environments with diverse expert demonstrations. In OPENDRAWER and CLOSEDRAWER, experts use varying approach paths to reach the handle; in MEATOFFGRILL and BLOCKPLACEMENT, the training dataset consists of distinct combinations of reaching and carrying trajectories; in PICKUPCUP, the robot must select between four discrete handles placed along the cup's rim to initiate a grasp. **Bottom row**: Testing environments featuring constraint-induced behavior shifts, showing one of the three evaluation variants tested for each task. Red barriers are introduced to obstruct trajectories associated with certain training modes (making them infeasible), or handles are removed to leave only a single feasible strategy.

ablation studies that each algorithmic component is necessary. We evaluate PDP across a suite of nine manipulation domains that exhibit high multimodality under identical initial conditions. The domains span both simulation and real-robot hardware and are designed to test strategy discovery, selection, and adaptation. The code and experiment configurations are available at https://github.com/Valarzz/pdp.

**Multimodal Dataset Construction.** We create a ***True multimodal dataset*** for each task, capturing both continuous variability within each mode and qualitatively distinct behaviors across modes. The benchmarks include single-stage domains (CLOSEDRAWER, OPENDRAWER, PICKUPCUP, OPENDOOR, OPENMICROWAVE, AVOIDING24, AVOIDING32), which require selecting of a single execution mode, and long-horizon domains (PLACEBLOCK, MEATOFFGRILL), where execution modes arise from the combinatorial composition of mode choices across independent reaching and carrying stages. More details are provided in Appendix B.1.

**Baselines.** For all domains, we compare PDP against eight state-of-the-art competitors spanning the main methodological families for this setting: standard Diffusion Policy (DP) (Chi et al., 2023), Behavior Cloning (BC) (Mandlekar et al., 2021), BC with Gaussian Mixtures (BC-GMM) (Calinon et al., 2007), Implicit Behavioral Cloning (IBC) (Florence et al., 2022), VQ-BeT (Lee et al., 2024), BESO (Reuss et al., 2023), Diffusion-ES (Diff-ES) (Yang et al., 2024), and ADPro (Li et al., 2025c). Together, these methods cover the major paradigms for modeling and adapting mul-

timodal robot behavior: generative, unimodal, explicit mixture, energy-based, vector-quantized, goal-conditioned, evolutionary-search, and test-time-adaptive approaches.

### 5.1. Simulation and Real-Robot Results

**Performance in Original Multimodal Domains.** We first evaluate each method in the original multimodal domains, before introducing any constraint shifts, to measure how well PDP preserves the behavior diversity present in the training data. As shown in Table 1, PDP achieves consistently high success rates across all tasks, including those with a large number of distinct behavior modes in the training data. This indicates that conditioning the diffusion process on a learned behavior latent code does not compromise fidelity in the original setting.

Standard DP, while generally outperforming unimodal BC, exhibits markedly lower success on tasks where distinct strategies induce highly divergent action geometries. Although DP can represent multimodal action distributions, unconditioned denoising struggles to reliably determine which strategy to execute, among a set of incompatible alternatives, often leading to noisy and inconsistent rollouts. By parameterizing behavior explicitly, PDP separates strategy selection from within-strategy denoising, simplifying learning and yielding substantially more stable execution. This effect is further illustrated by the trajectory visualizations in Appendix C.2: while non-diffusion baselines struggle to capture the underlying multimodal structure and DP fails to consistently steer specific modes, PDP robustly reconstructs

*Table 1.* Success rate (%) on **original** simulation scenes before constraint shifts (mean±std over 3 random seeds). The best performance in each row is bolded.

| Task | PDP | DP | BC-GMM | BC | IBC | ADPro | Diff-ES | VQ-BeT | BESO |
|---|---|---|---|---|---|---|---|---|---|
| CLOSEDRAWER | **100.0 ± 0.0** | 88.3 ± 3.8 | 70.8 ± 10.1 | 23.3 ± 1.4 | 9.2 ± 6.3 | 91.7 ± 1.7 | 99.2 ± 0.8 | **100.0 ± 0.0** | 92.5 ± 2.5 |
| PLACEBLOCK | **97.5 ± 2.5** | 60.0 ± 6.6 | 5.0 ± 2.5 | 0.8 ± 1.4 | 0.8 ± 1.4 | 15.0 ± 3.8 | 80.0 ± 0.0 | 77.5 ± 0.0 | 29.2 ± 3.0 |
| MEATOFFGRILL | **73.3 ± 2.9** | 52.5 ± 8.7 | 52.5 ± 0.0 | 55.0 ± 0.0 | 0.0 ± 0.0 | 0.0 ± 0.0 | 60.0 ± 0.0 | 0.0 ± 0.0 | 30.8 ± 7.1 |
| PICKUPCUP | **90.0 ± 5.0** | 15.8 ± 3.8 | 89.2 ± 1.4 | 41.7 ± 7.2 | 0.8 ± 1.4 | 52.5 ± 4.3 | 40.0 ± 0.0 | 79.2 ± 1.7 | 9.2 ± 4.2 |

*Table 2.* Success rate (%) on **constraint-shifted** simulation scenes for eight manipulation domains, comparing nine methods (mean±std over 3 seeds). Scenes 1–2 introduce constraint shifts that make all but one training mode infeasible. Scene 3 is a **zero-mode-feasible** setting: the new constraints make *all* training modes infeasible, and a single new demonstration is provided for adaptation. The best performance in each row is shown in bold.

| Task | Scene | PDP | DP | BC-GMM | BC | IBC | ADPro | Diff-ES | VQ-BeT | BESO |
|---|---|---|---|---|---|---|---|---|---|---|
| CLOSEDRAWER | 1 | **100.0 ± 0.0** | **100.0 ± 0.0** | 0.0 ± 0.0 | 0.0 ± 0.0 | 0.0 ± 0.0 | **100.0 ± 0.0** | 99.2 ± 0.7 | **100.0 ± 0.0** | 92.5 ± 2.5 |
| | 2 | **100.0 ± 0.0** | 88.3 ± 14.2 | 0.0 ± 0.0 | 82.5 ± 4.3 | 0.0 ± 0.0 | 96.7 ± 1.8 | 34.2 ± 2.5 | **100.0 ± 0.0** | 97.5 ± 2.5 |
| | 3 | **100.0 ± 0.0** | 40.0 ± 13.0 | 0.0 ± 0.0 | 2.5 ± 2.5 | 0.0 ± 0.0 | 0.0 ± 0.0 | 1.7 ± 0.7 | **100.0 ± 0.0** | **100.0 ± 0.0** |
| PLACEBLOCK | 1 | **95.0 ± 2.5** | 0.0 ± 0.0 | 0.0 ± 0.0 | 12.5 ± 3.8 | 0.0 ± 0.0 | 0.0 ± 0.0 | 14.2 ± 4.8 | 90.0 ± 2.5 | 0.0 ± 0.0 |
| | 2 | **95.8 ± 1.4** | 0.0 ± 0.0 | 0.0 ± 0.0 | 2.5 ± 2.5 | 0.0 ± 0.0 | 0.0 ± 0.0 | 13.3 ± 5.4 | 92.2 ± 0.9 | 22.5 ± 15.9 |
| | 3 | **86.7 ± 10.1** | 2.5 ± 2.5 | 0.0 ± 0.0 | 5.0 ± 2.5 | 0.0 ± 0.0 | 0.0 ± 0.0 | 13.3 ± 5.4 | 76.0 ± 3.1 | 16.2 ± 11.5 |
| MEATOFFGRILL | 1 | **84.2 ± 1.4** | 0.0 ± 0.0 | 0.0 ± 0.0 | 80.0 ± 5.0 | 0.0 ± 0.0 | 0.0 ± 0.0 | 0.0 ± 0.0 | 0.0 ± 0.0 | 0.0 ± 0.0 |
| | 2 | **86.7 ± 5.2** | 0.0 ± 0.0 | 0.0 ± 0.0 | 65.0 ± 3.8 | 0.0 ± 0.0 | 0.8 ± 0.7 | 0.0 ± 0.0 | 7.5 ± 2.5 | 0.0 ± 0.0 |
| | 3 | **91.7 ± 2.9** | 2.5 ± 2.5 | 0.0 ± 0.0 | 0.0 ± 0.0 | 0.0 ± 0.0 | 0.8 ± 0.7 | 2.5 ± 2.5 | 0.0 ± 0.0 | 0.0 ± 0.0 |
| PICKUPCUP | 1 | **95.8 ± 1.4** | 7.5 ± 13.0 | 92.5 ± 2.5 | 2.5 ± 2.5 | 0.0 ± 0.0 | 0.0 ± 0.0 | 20.0 ± 4.1 | 85.0 ± 2.5 | 0.0 ± 0.0 |
| | 2 | **85.0 ± 2.5** | 0.8 ± 1.4 | 72.5 ± 4.3 | 4.2 ± 2.9 | 0.0 ± 0.0 | 10.8 ± 4.9 | 20.0 ± 3.5 | 55.0 ± 5.0 | 22.5 ± 15.9 |
| | 3 | **82.5 ± 4.3** | 15.8 ± 21.0 | 0.0 ± 0.0 | 22.5 ± 4.3 | 0.0 ± 0.0 | 3.3 ± 1.4 | 20.0 ± 2.5 | 70.0 ± 2.5 | 16.2 ± 11.5 |
| *Overall Avg.* | | **92.0** | 21.5 | 13.8 | 23.3 | 0.0 | 17.7 | 19.9 | 64.6 | 30.6 |

intended trajectories with minimal intra-mode noise.

**Adaptation to Constraint-Induced Shifts.** We next evaluate behavior generalization under constraint-induced shifts, where environmental changes make most behavior modes covered in the training data infeasible while leaving the task goal unchanged. Each domain is tested under two evaluation regimes: (i) **Existing Mode Fitting**, with two *constraint-shifted scenes* (Scenes 1–2) in which obstacles or affordance removals make all but one training mode infeasible; and (ii) **Novel Behavior Discovery**, a *zero-mode-feasible* setting (Scene 3) where no training mode remains valid and success requires discovering a qualitatively new behavior. Fig. 4 depicts each domain's original scene together with one representative constraint shift; the remaining shift variants and task specifications are detailed in Appendix B.3. Due to space constraints, Table 2 reports results on four representative domains, and we defer the complete results across all eight domains to Appendix B.2.

In all constraint-shifted scenes, we provide a single successful demonstration $\tilde{\tau}$ and evaluate each method's ability to adapt from it. PDP aims to enable such adaptation without retraining. PDP adapts by optimizing only the behavior latent code $z$ via test-time fitting; DP performs optimization in its high-dimensional noise space; BC fine-tunes policy weights with a small learning rate; BC-GMM selects mixture components via posterior likelihood; IBC biases inference toward $\tilde{\tau}$; Diff-ES uses the DTW distance to $\tilde{\tau}$ as

its evolutionary-search objective; ADPro guides denoising with the gradient of a soft-DTW objective relative to $\tilde{\tau}$; and BESO and VQ-BeT condition on waypoints extracted from $\tilde{\tau}$ as goals. Additional details are provided in Appendix B.4.

As shown in Table 2, PDP maintains high success across all constraint-shifted evaluations, while all baseline methods exhibit severe performance degradation.

In the **Existing Mode Fitting** setting, PDP succeeds nearly perfectly across all tasks, demonstrating that the learned latent space provides a stable control handle for steering the policy. In contrast, standard DP and other frozen-model baselines fail to consistently converge, despite the correct strategy being present in the training data. Although BC achieves moderate success by explicitly fine-tuning its network parameters using the new demonstration, this form of adaptation relies on parameter updating rather than the model's intrinsic capacity for controllable behavior selection.

In the **Novel Behavior Discovery** setting, PDP remains the only method that consistently succeeds across domains, indicating that its geometry-aligned latent space defines a behavior manifold that can be navigated to synthesize feasible new strategies. All baselines fail in this regime, including BC, despite being allowed to update its weights using the new demonstration, suggesting that local parameter adaptation alone is insufficient to synthesize behaviors absent from the training data without an explicitly structured behavior space.

*Table 3.* Real-robot OPENDRAWER results (successes / 5 trials).

| Method | Original | Scene 1 | Scene 2 | Scene 3 |
| --- | --- | --- | --- | --- |
| PDP | **5/5** | **5/5** | **5/5** | **5/5** |
| DP | 5/5 | 3/5 | 1/5 | 2/5 |

*Table 4.* Effect of latent integration strategy in the denoiser.

| Task | Isometric | Unaligned | Discrete |
| --- | --- | --- | --- |
| CLOSEDRAWER | **100.0 ± 0.0** | 11.7 ± 20.2 | 0.0 ± 0.0 |
| PICKUPCUP | **82.5 ± 0.0** | 62.5 ± 6.6 | 0.0 ± 0.0 |

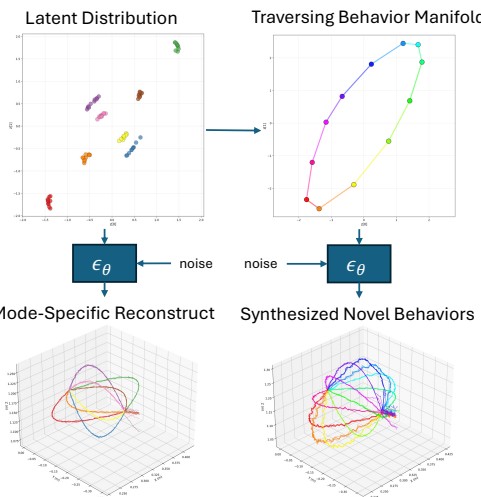

*Figure 5.* **Generalization via latent-space navigation.** The learned behavior latent space organizes demonstrations into compact clusters, with Euclidean distances reflecting trajectory similarity. Navigating the manifold by smoothly interpolating between latent clusters allows the denoiser $\epsilon_\theta$ to generate discover novel behaviors between demonstrated modes.

**Real-Robot Robustness to Stochasticity.**[1] In addition to simulation experiments, we evaluate PDP on a real-world manipulation task using a Franka Emika Panda robot arm. The task is OPENDRAWER, which requires the robot to reach a drawer handle, establish contact, and pull the drawer open to a target configuration. Real-world demonstrations in this task are characterized by non-smooth trajectories and significant intra-mode variability (see Appendix B.1). Despite this, Table 3 shows that PDP achieves a 100% success rate (5/5) across all scenes, including in the challenging zero-mode adaptation setting. Although standard DP matches PDP's performance in the original scene, its success drops to as low as 1/5 (20% success rate) under constraint shifts. These results demonstrate that PDP's learned behavior latent space is robust to real-world execution noise and provides a more stable handle for test-time policy steering.

**5.2. Generalization via Behavior-Space Navigation**

We provide a qualitative analysis of the learned behavior manifold to support our claims regarding geometric regularity and the resulting steerability of the policy. Fig. 5 depicts the behavior embedding and synthesis process on CLOSEDRAWER. As expected from the metric alignment

enforced by $\mathcal{L}_{\text{geo}}$, the latent space is organized into compact, well-separated clusters corresponding to the demonstration modes. To test the manifold's capacity for novel behavior synthesis, we generate trajectories by conditioning the policy on an interpolated latent code $z(\lambda)$, where $\lambda$ indexes positions along a chosen path in the latent space, without any additional training. As $z(\lambda)$ traverses the latent manifold along elliptical sweeps or linear paths between clusters, the resulting end-effector (EE) trajectories exhibit smooth semantic transitions. This behavior empirically supports our dynamical-systems analysis in Appendix A.4: by navigating between modes in a low-dimensional coordinate chart, we shift the equilibrium of the denoiser's probability flow to synthesize intermediate, physically plausible strategies that are not exact replicas of the training data. Beyond the example shown here, Appendix C.3 provides additional results in different domains, investigating other types of paths that can be followed while navigating the latent space.

**5.3. Ablation Studies**

We conduct a series of ablation studies on the CLOSE-DRAWER and PICKUPCUP domains to evaluate the contribution of our core architectural choices. Unless otherwise specified, all variants are evaluated on the *zero-mode-feasible* variant (Scene 3) to test their impact on novel behavior adaptation.

**Impact of Geometry Alignment & Latent Continuity.** We first evaluate the need for metric alignment and for a continuous behavior manifold by comparing three variants defined by their latent construction: (i) **Isometric**, where PDP is conditioned on $\bar{z}$ from an encoder trained with $\mathcal{L}_{\text{embed}}$); (ii) **Unaligned**, where PDP is conditioned on $\bar{z}$ from an encoder trained with $\beta_{\text{geo}} = 0$; and (iii) **Discrete**, where PDP is conditioned on a categorical one-hot embedding.

Table 4 shows that removing geometric alignment causes a severe drop in adaptation performance, while the Discrete variant fails entirely. The Isometric latent space, by contrast, organizes behaviors according to physical trajectory similarity, so interpolation yields smooth, intermediate strategies. The Unaligned space, on the other hand, is geometrically warped. Effective test-time adaptation requires both a geometry-aligned latent and a continuous behavior manifold; without either, gradient-based latent fitting cannot reliably synthesize novel behaviors.

**Importance of Global Modulation for Latent Integration.** We now investigate how the method used to integrate the

---

[1]Videos of all real-robot experiments are available at https://sites.google.com/view/parameterized-dp.

*Table 5.* Effect of behavior latent integration on adaptation performance.

| Task | Global Modulation | Concatenation | Unconditioned |
|------|-------------------|---------------|---------------|
| CLOSEDRAWER | **100.0 ± 0.0** | 82.5 ± 4.3 | 40.0 ± 13.0 |
| PICKUPCUP | **82.5 ± 0.0** | 25.0 ± 0.0 | 15.8 ± 21.0 |

*Table 6.* Effect of initialization strategy & latent optimization steps.

| Task | Init | 0 steps | 10 steps | 100 steps |
|------|------|---------|----------|-----------|
| CLOSEDRAWER | Warm | **98.3 ± 1.4** | **100.0 ± 0.0** | **100.0 ± 0.0** |
|             | Rand | 0.0 ± 0.0 | 0.0 ± 0.0 | 2.5 ± 2.5 |
| PICKUPCUP | Warm | **61.7 ± 6.3** | **81.7 ± 5.2** | **82.5 ± 0.0** |
|           | Rand | 3.3 ± 3.8 | 15.8 ± 6.3 | 53.3 ± 10.1 |

behavior signal affects the denoiser's ability to resolve mode ambiguity. We compare: (i) **Global Modulation** (PDP using affine feature-map modulation), (ii) **Concatenation** (PDP with $z$ appended to the input), and (iii) **Unconditioned** (standard DP without a latent $z$).

The results in Table 5 show that our global modulation architecture yields substantially higher adaptation success rates than naive concatenation, while also reducing diffusion training loss by nearly an order of magnitude (Appendix C.1). This stark contrast supports our claim that deep, explicit latent integration simplifies diffusion from a global mixture-modeling problem into a mode-specific denoising process, facilitating training and enabling the denoiser to more reliably track the target behavior.

**Importance of Semantic Warm-Starts.** Finally, we evaluate the role of the learned encoder $E_\phi$ in the test-time fitting process. We compare two strategies for initializing latent optimization: using the encoder prediction $z_0 = E_\phi(\tilde{\tau})$ (**Warm**) and using a random Gaussian sample (**Rand**). Table 6 shows that the **Warm** initialization achieves high success with as few as 10 optimization steps. In contrast, random initialization fails to reach viable strategies even after 100 steps. These results show that our geometry-aligned encoder provides a critical semantic initialization, placing the optimization in a well-conditioned region of the behavior manifold.

## 6. Conclusion

We propose Parameterized Diffusion Policy, a framework that learns a continuous and geometry-aligned behavior space for steering diffusion policies. By structuring a latent manifold so that distances between representations reflect the semantic similarity between physical trajectories, we transform diffusion from a mechanism for stochastic diversity into a precise tool for behavior steering. Our approach enables smooth interpolation between known strategies and efficient adaptation to novel constraints. We demonstrate that PDP significantly improves success rates on complex multimodal benchmarks in both simulated and real-robot experiments, particularly in scenarios requiring the synthesis of entirely new behaviors.

## Acknowledgments

We thank Scott Niekum for early discussions related to this project and Zilai Zeng for discussions on the real-robot setup. This work used computational resources and services provided by the Unity Research Computing Platform. We also thank the anonymous reviewers for their comments and suggestions, which helped strengthen and clarify this work.

## Impact Statement

This paper presents work whose goal is to improve the controllability and adaptability of diffusion-based robot policies under changing task constraints. By learning a geometry-aligned latent behavior space, the proposed method provides an explicit interface for adapting to new constraints, which may help make robot policies more reliable and data-efficient in settings where feasible behaviors change at deployment time. These benefits should be interpreted within the limits of the method: adaptation relies on a successful demonstration, and the adapted behavior is expected to lie within or near the learned behavior space induced by existing demonstrations.

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

# A. Algorithms and Model Architecture

## A.1. DTW and Soft-DTW Formulation

To establish a geometry-aligned behavior space, we require a distance metric $\delta^X$ that is invariant to temporal variations and execution speeds. We use Dynamic Time Warping (DTW) and its differentiable counterpart, Soft-DTW, to compare behavior traces $X(\tau_i)$ and $X(\tau_j)$. Let these traces be represented as sequences $A = (a_1, \ldots, a_n)$ and $B = (b_1, \ldots, b_m)$.

**Dynamic Time Warping (DTW):** DTW seeks an optimal alignment between $A$ and $B$ by minimizing the cumulative cost over a cost matrix $D \in \mathbb{R}^{n \times m}$, where $D_{i,j} = \Delta(a_i, b_j)$ represents the local distance between points. An alignment is defined by a binary warping path (alignment matrix) $\mathbf{A} \in \{0,1\}^{n \times m}$, where $\mathbf{A}_{i,j} = 1$ if $a_i$ is matched to $b_j$. The set of all valid paths $\mathcal{A}_{n,m}$ must satisfy boundary, continuity, and monotonicity constraints. The standard DTW distance is defined as:

$$DTW(A, B) = \min_{\mathbf{A} \in \mathcal{A}_{n,m}} \langle \mathbf{A}, D \rangle. \tag{13}$$

This is computed efficiently via dynamic programming with the recurrence:

$$r_{i,j} = D_{i,j} + \min\{r_{i-1,j}, r_{i,j-1}, r_{i-1,j-1}\}. \tag{14}$$

**Soft-DTW Formulation:** The $\min$ operator in standard DTW is non-differentiable, preventing backpropagation through the geometry loss $\mathcal{L}_{geo}$. Following Cuturi & Blondel (2017), we replace the hard $\min$ with a smoothed $\text{softmin}_\gamma$ operator with a parameter $\gamma > 0$:

$$softDTW_\gamma(A, B) = -\gamma \log \sum_{\mathbf{A} \in \mathcal{A}_{n,m}} \exp\left(-\frac{\langle \mathbf{A}, D \rangle}{\gamma}\right). \tag{15}$$

As $\gamma \to 0$, $softDTW_\gamma$ converges to the standard DTW distance. The value can be computed in $O(nm)$ time using the following DP recurrence:

$$r_{i,j} = D_{i,j} + \text{softmin}_\gamma\{r_{i-1,j}, r_{i,j-1}, r_{i-1,j-1}\}, \tag{16}$$

where $\text{softmin}_\gamma(x_1, \ldots, x_k) = -\gamma \log \sum_{i=1}^{k} \exp(-x_i/\gamma)$.

**Gradient Computation:** The primary advantage of Soft-DTW is its differentiability with respect to the input sequences. The gradient with respect to the cost matrix $D$ (and thus the behavior latent $z$ via the trajectory features) is given by:

$$\nabla_D softDTW_\gamma(A, B) = E = [e_{i,j}], \tag{17}$$

where $E$ is the expected alignment matrix under the Gibbs distribution $P(\mathbf{A}) \propto \exp(-\langle \mathbf{A}, D \rangle/\gamma)$. This allows the PDP framework to anchor the Euclidean geometry of the behavior manifold $z$ to the physical similarity of trajectories by minimizing the alignment-based distance $\delta_{ij}^X$.

---

**Algorithm 1** Soft-DTW Forward Pass

---

1: **Input:** Cost matrix $D \in \mathbb{R}^{n \times m}$, smoothing parameter $\gamma$.
2: $R \in \mathbb{R}^{(n+1) \times (m+1)} \leftarrow \infty$
3: $R_{0,0} \leftarrow 0$
4: **for** $i = 1$ to $n$ **do**
5:     **for** $j = 1$ to $m$ **do**
6:         $r_{i,j} = D_{i,j} + \text{softmin}_\gamma\{R_{i-1,j}, R_{i,j-1}, R_{i-1,j-1}\}$
7:     **end for**
8: **end for**
9: **Output:** $R_{n,m}$

---

## A.2. Joint Training of the Behavior Latent and Policy

The training of PDP requires balancing two distinct objectives: establishing a structured, geometry-aligned latent manifold $z$ and learning a denoiser $\epsilon_\theta$ that can accurately map these latents to high-dimensional action sequences. A naive joint

optimization could allow the diffusion loss—which is primarily a mode-fitting objective—to distort the metric-preserving properties of the latent space enforced by the soft-DTW geometry loss $\mathcal{L}_{geo}$.

To prevent this representation collapse, we use an alternating optimization scheme. In the first phase, we refine the trajectory encoder $E_\phi$ and symmetric decoder $D_\psi$ to ensure that the latent space $z$ preserves behavioral information and captures semantic similarities between physical trajectories. In the second phase, we optimize the denoiser parameters $\theta$ by conditioning on the latent $z$ produced by the encoder. Crucially, we apply a stop-gradient operator $sg(\cdot)$ to the behavior latent during this phase, ensuring $z$ remains a fixed, reliable control handle for the diffusion process. We provide the detailed procedural breakdown of this joint training scheme in Algorithm 2.

---

**Algorithm 2** Joint Training of Parameterized Diffusion Policy (PDP)

---

1: **Initialize:** Encoder $E_\phi$, Decoder $D_\psi$, and Denoiser $\epsilon_\theta$.
2: **while** not converged **do**
3:   Sample batch of demonstrations $\{\tau_i, \tau_j\}$ from dataset.
4:   **Step 1: Embedding Refinement**
5:   Compute physical distances $\delta_{ij}^X = softDTW_\gamma(X(\tau_i), X(\tau_j))$.
6:   Map to latents $z = \mu_\phi(\tau) + \sigma_\phi(\tau) \odot \xi$ via trajectory encoder.
7:   Update $(\phi, \psi)$ via $\nabla_{\phi,\psi}(\mathcal{L}_{rec} + \beta_{KL}\mathcal{L}_{KL} + \beta_{geo}\mathcal{L}_{geo})$.
8:   **Step 2: Denoiser Optimization**
9:   Compute fixed behavior latent $\bar{z} = sg(\mu_\phi(\tau))$.
10:   Update $\theta$ via $\nabla_\theta \mathbb{E}_{\tau,t,k,\epsilon}[\|\epsilon - \epsilon_\theta(A_t^k, k, c_t, \bar{z})\|_2^2]$.
11: **end while**

---

## A.3. Test-Time Latent Adaptation via Gradient Descent

A core advantage of PDP is its ability to perform fast, data-light adaptation to novel environmental constraints without updating millions of network weights. Standard diffusion policies often rely on stochastic sampling to discover feasible behaviors, which is ill-conditioned under significant constraint shifts where successful trajectories occupy a small fraction of the total noise space. In contrast, PDP treats the behavior manifold as a structured search space, replacing "luck-based" sampling with stable, gradient-based optimization in $z$.

The adaptation process begins with a "semantic warm-start," where we use the frozen encoder $E_\phi$ to map a target demonstration $\tilde{\tau}$ (or a partial behavior trace) to an initial latent $z_0$ code. This places the optimization in the correct region of the behavior manifold, significantly improving convergence stability compared to random initialization. We then iteratively refine this latent representation by minimizing a diffusion-consistent fitting objective $\mathcal{L}_{fit}$, which identifies the behavior mode that maximizes the likelihood of the target actions under the frozen denoiser. The complete test-time latent adaptation procedure is formalized in Algorithm 3.

---

**Algorithm 3** Test-Time Latent Adaptation

---

1: **Input:** Target demonstration $\tilde{\tau}$, frozen models $\{E_\phi, \epsilon_\theta\}$.
2: **Initialization:** $z_0 = \mu_\phi(X(\tilde{\tau}))$ {Semantic Warm-start}
3: **for** step $s = 1$ to $S$ **do**
4:   Sample target action chunks $\tilde{A}_t$ and context $\tilde{c}_t$.
5:   Compute $\mathcal{L}_{fit}(z) = \mathbb{E}_{k,\epsilon}[\|\epsilon - \epsilon_\theta(\tilde{A}_t^k, k, \tilde{c}_t, z)\|_2^2] + \lambda\|z\|_2^2$.
6:   Update latent: $z \leftarrow z - \eta\nabla_z\mathcal{L}_{fit}(z)$.
7: **end for**
8: **Output:** Optimized $z^*$ for execution rollouts.

---

## A.4. Analysis of Controllability and Generalization

This section provides an analysis of why PDP improves over standard DP in the constraint-induced behavior shift setting studied in this paper (Fig. 1). The key distinction is between (i) *representing* multimodal behaviors in offline demonstrations, and (ii) *steering* a policy toward a specific feasible strategy when constraints shift. We show that DP addresses (i) better than regression-based BC, but its default control interface—high-dimensional sampling noise—is ill-conditioned (ii). PDP

resolves this by exposing a low-dimensional behavior latent $z$ whose geometry is explicitly aligned with physical trajectory similarity via soft-DTW, making both training and test-time steering well-conditioned.

**Diffusion policies can represent multimodal behaviors while regression BC collapses them.** Consider a context $c_t$ (e.g., a history encoder output) and an action chunk $A_t$ of horizon $H$. In our *True multimodal* datasets, the expert conditional distribution is inherently multi-valued:

$$p_{\text{train}}(A_t \mid c_t) = \sum_{m=1}^{M} p(m \mid c_t)\, p(A_t \mid c_t, m), \tag{18}$$

where $m$ indexes distinct trajectory modes (e.g., different approach paths or grasp affordances). A standard regression BC objective (e.g., squared error on action chunks) learns a deterministic predictor $\pi_\theta(c_t)$ that minimizes $\mathbb{E}[\|A_t - \pi_\theta(c_t)\|_2^2]$. The population minimizer is the conditional mean $\pi^*(c_t) = \mathbb{E}[A_t \mid c_t]$, which averages across modes in Eq. (18). When modes are geometrically incompatible, this mean corresponds to an "in-between" trajectory that may be physically infeasible. In contrast, diffusion policies model a *conditional distribution* over chunks and can sample distinct behaviors from different noise seeds, enabling multi-strategy rollouts when the data are truly multimodal.

At the same time, recent analysis of generative control policies (Pan et al., 2025a) emphasizes that on many popular BC benchmarks, performance gains often arise not from learned multi-modality per se, but from the combination of stochasticity injection during training and supervised iterative computation during inference. Our setting is complementary: we intentionally construct benchmarks where multiple distinct strategies exist under near-identical initial conditions, so explicit mode handling is essential—and we further require a *controllable* interface to reliably select or synthesize feasible strategies under constraint shifts.

**Mode disambiguation yields training relief.** DP trains a denoiser to predict diffusion noise under the forward process

$$A_t^k = \sqrt{\bar{\alpha}_k}\, A_t + \sqrt{1 - \bar{\alpha}_k}\, \epsilon, \qquad \epsilon \sim \mathcal{N}(0, I), \tag{19}$$

using an MSE noise-prediction loss (9). For notational brevity, let $x \triangleq (A_t^k, k, c_t)$. Under squared loss, the Bayes-optimal predictor is the conditional mean $\epsilon^*(x) = \mathbb{E}[\epsilon \mid x]$, and the minimum achievable risk is the conditional variance $\mathbb{E}[\text{Var}(\epsilon \mid x)]$. When $p(A_t \mid c_t)$ is multimodal, $x$ alone does not identify the underlying mode $m$, so $\epsilon$ remains highly uncertain given $x$, leading to a harder regression problem and unstable "averaged" denoising directions.

PDP introduces a behavior latent $z$ to separate *mode selection* from *within-mode denoising*. Writing the conditional risk with $z$ gives the variance decomposition (law of total variance):

$$\text{Var}(\epsilon \mid x) = \mathbb{E}_z[\text{Var}(\epsilon \mid x, z)] + \text{Var}_z(\mathbb{E}[\epsilon \mid x, z]). \tag{20}$$

The first term $\mathbb{E}_z[\text{Var}(\epsilon \mid x, z)]$ represents *residual within-mode uncertainty* (intra-mode noise, partial observability, and diffusion corruption) after a behavior is specified. The second term,

$$\Delta_{\text{mode}}(x) \triangleq \text{Var}_z(\mathbb{E}[\epsilon \mid x, z]), \tag{21}$$

is the *structural between-mode variance*: it measures how much the optimal denoising target changes when we vary the desired behavior. This is precisely the "variance gap" paid by an unconditioned DP when it must implicitly average over multiple incompatible strategies.

Importantly, in PDP this decomposition is not merely formal: $z$ is trained to have a specific semantic meaning. Our encoder is optimized with a geometry loss (7) enforcing

$$\|\bar{z}(\tau_i) - \bar{z}(\tau_j)\|_2 \approx \kappa\, d_{\text{softDTW}}(X(\tau_i), X(\tau_j)), \tag{22}$$

where $X(\tau)$ is a behavior trace capturing the physical trajectory geometry. Since distinct modes in our datasets correspond to large changes in the trajectory trace (e.g., different approach paths or contact patterns), this alignment organizes $z$ into compact, well-separated regions that correlate with mode identity. Consequently, conditioning on $z$ makes $p(m \mid x, z)$ sharply peaked (Dirac-like in the idealized limit), so the denoiser learns a simpler *mode-specific* denoising problem rather than a global mixture. Empirically, this improved conditioning of the training process is reflected as a large reduction in

diffusion training loss for latent-conditioned models versus unconditioned or weakly conditioned baselines (Sec. 5.3), and yields substantially improved reliability in high-mode tasks (Table 1).

A further interpretation follows from rearranging Eq. (19):

$$\epsilon = \frac{A_t^k - \sqrt{\bar{\alpha}_k}\, A_t}{\sqrt{1 - \bar{\alpha}_k}}. \tag{23}$$

Conditioned on $(A_t^k, k, c_t)$, uncertainty in $\epsilon$ is induced by uncertainty in the clean chunk $A_t$. Since Eq. (23) is affine in $A_t$, we have that

$$\text{Var}(\epsilon \mid x) = \frac{\bar{\alpha}_k}{1 - \bar{\alpha}_k}\, \text{Var}(A_t \mid x), \qquad \text{Var}(\epsilon \mid x, z) = \frac{\bar{\alpha}_k}{1 - \bar{\alpha}_k}\, \text{Var}(A_t \mid x, z). \tag{24}$$

Thus, $\Delta_{\text{mode}}(x)$ in Eq. (21) directly corresponds (up to the same diffusion-dependent scaling) to variance in the conditional mean of the clean action chunk across behavior latents, i.e., to the separation between strategy manifolds. PDP reduces this structural variance by making $z$ an explicit index of the trajectory geometry.

**Low-dimensional steering enables controllable adaptation under constraint shifts.**   While DP can represent multi-modality, its default control interface for selecting behaviors is the sampling noise $\epsilon \in \mathbb{R}^{Hd_a}$ (or equivalently the initial latent of the reverse process). For multimodal tasks, each strategy corresponds to a basin of attraction in this high-dimensional space. Under constraint-induced shifts, feasible behaviors may occupy a small and hard-to-reach subset of noise space, making both random sampling and direct noise-space optimization ill-conditioned.

PDP replaces this implicit interface with an explicit low-dimensional control handle $z \in \mathbb{R}^{d_z}$, where $d_z \ll Hd_a$, and where distances in $z$ are aligned with physical trajectory similarity (Eq. (22)). At test time, we fit $z$ by minimizing the diffusion-consistent objective (10):

$$L_{\text{fit}}(z; \tilde{\tau}) = \mathbb{E}_{t,k,\epsilon}\left[\|\epsilon - \epsilon_\theta(\tilde{A}_t^k, k, \tilde{c}_t, z)\|_2^2\right] + \lambda\|z\|_2^2, \tag{25}$$

initialized from a *semantic warm-start* $z_0 = \mu_\phi(X(\tilde{\tau}))$. The first term identifies the behavior latent under which the frozen denoiser assigns high likelihood to the demonstrated actions, while the quadratic penalty is consistent with the Gaussian prior induced by the KL term in (5) and keeps the optimization within the support of the learned manifold. Because $z$ is a geometry-aligned coordinate chart, gradient descent in $z$ corresponds to coherent movement in trajectory space: small changes in $z$ induce small, predictable changes in the generated behavior. This enables both (i) reliable selection of the unique feasible training mode (Scenes 1–2), and (ii) navigation to new regions of the manifold to discover novel strategies when no training mode is feasible (Scene 3), without updating policy weights.

**Generalization as navigable interpolation in behavior space.**   Finally, Eq. (22) provides a principled explanation for why interpolation in $z$ yields semantically meaningful behavior interpolation. If $z(\lambda) = (1 - \lambda)z_i + \lambda z_j$ lies between two mode regions, then its Euclidean position corresponds to an intermediate soft-DTW distance in the behavior-trace metric, so the policy is biased toward synthesizing trajectories that are physically intermediate between the two endpoint strategies. This effect is visible in our latent traversal visualizations (Fig. 5): in an isometric manifold, midpoint latents generate smooth intermediate paths, whereas an unaligned latent collapses to unrelated modes.

Overall, PDP improves controllability and generalization by (i) providing *training relief* through mode disambiguation, reducing the inter-mode variance term $\Delta_{\text{mode}}(x)$ that otherwise burdens the denoiser, and (ii) providing *low-dimensional steering* through a geometry-aligned latent $z$ that converts diffusion from noise-driven diversity into a stable and optimizable mechanism for behavior selection, interpolation, and rapid adaptation under constraint-induced behavior shifts.

## B. Domain Specifications and Implementation Details

### B.1. Manipulation Task Descriptions and Multimodal Dataset Collection

#### B.1.1. SIMULATION SETUP

To evaluate the efficacy of PDP in handling high-dimensional, multimodal robotic control, we use a benchmark suite of manipulation domains from RLBench, including CLOSEDRAWER, OPENDRAWER, PICKUPCUP, PLACEBLOCK, and MEATOFFGRILL. RLBench is a standardized robotic manipulation benchmark built on top of a physics-based simulator with

a fixed robot embodiment and a large collection of goal-conditioned tasks. Beyond RLBench, we further evaluate on four other domains drawn from three complementary benchmarks (Figure 6): OPENDOOR from robomimic (Mandlekar et al., 2021), OPENMICROWAVE from the Franka Kitchen environment (Gupta et al., 2019), and AVOIDING24 and AVOIDING32 from the D3IL benchmark (Jia et al., 2024). Spanning distinct simulation engines, robot embodiments, and task structures, these benchmarks together provide a comprehensive testbed for multimodal behavior learning and adaptation. Each task is formulated as a finite-horizon Markov Decision Process (MDP) defined by $(\mathcal{S}, \mathcal{A}, \mathcal{P}, r, \gamma)$, where the state $s_t \in \mathcal{S}$ encodes the instantaneous configuration of the robot and its environment, including task-relevant object poses and robot kinematics, and actions correspond to continuous end-effector control executed through stochastic transition dynamics $\mathcal{P}$. For all domains, we introduce targeted modifications to the original task setups to make them better suited for controlled multimodal data collection, including the insertion of obstacles between the gripper and task-relevant objects or targets, as well as the restriction of feasible grasping affordances. These modifications are designed to induce multiple distinct but valid strategies under identical initial conditions. We provide detailed descriptions of all task setups below.

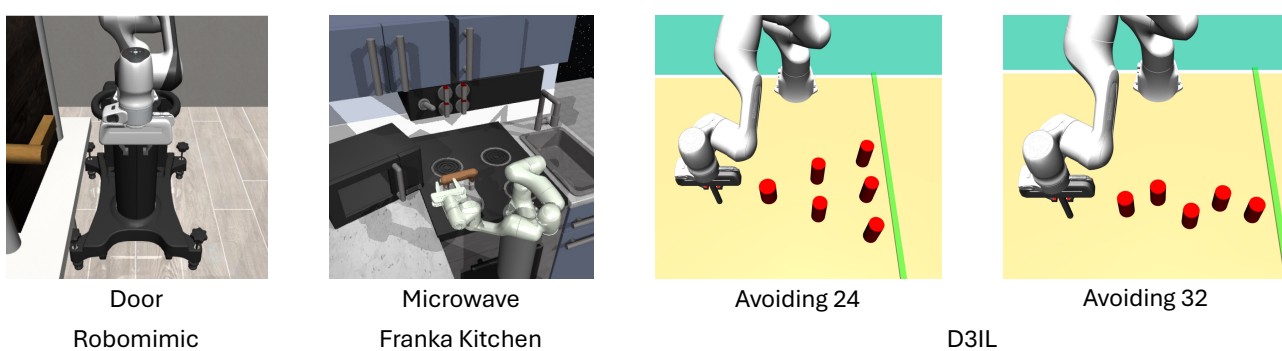

| Door | Microwave | Avoiding 24 | Avoiding 32 |
| Robomimic | Franka Kitchen | D3IL | |

*Figure 6.* The four other manipulation domains we evaluate on, aside from those shown in the main text: OPENDOOR (robomimic), OPENMICROWAVE (Franka Kitchen), and AVOIDING24 and AVOIDING32 (D3IL).

- **CloseDrawer.** The objective of CLOSEDRAWER is to close an open drawer by pushing its handle to a target closed configuration. To induce multimodality, we introduce a static obstacle positioned between the gripper and the drawer handle, blocking direct straight-line access. As a result, the robot must execute a curved approach trajectory that circumvents the obstacle before making contact with the handle and applying the closing motion. Multiple feasible strategies arise depending on how the gripper navigates around the obstacle. The episode is considered successful when the drawer reaches the closed threshold without contacting the obstacle; any collision with the obstacle or failure to close the drawer results in episode failure.

- **PickUpCup.** In PICKUPCUP, the robot is required to grasp and lift a cup from a table. Rather than inducing multimodality through reaching trajectories, this task emphasizes multimodality in grasp selection. We manually define four discrete grasping affordances located at angular positions of $0°$, $90°$, $180°$, and $270°$ around the rim of the cup, corresponding to four distinct grasping strategies. The gripper is restricted to grasping at these predefined locations; grasps at any other position are considered invalid. Since the default RLBench cup does not provide such discrete handles, we explicitly specify these grasping points to simulate the intended affordance structure. An episode succeeds only if the cup is grasped at one of the valid locations and lifted to the target height; grasping at an incorrect location or dropping the cup leads to failure.

- **PlaceBlock.** The PLACEBLOCK task requires the robot to pick up a block and place it at a designated target location. To promote multimodal behavior, we insert two obstacles into the scene: one between the gripper and the block, and another between the block's initial position and the target placement region. This setup forces the robot to choose among multiple distinct reaching paths to grasp the block, as well as different transport trajectories to carry it to the target while avoiding collisions. The episode is successful if the block is placed within the target region without contacting any obstacle; collisions during either the reaching or placing phase result in failure.

- **MeatOffGrill.** In MEATOFFGRILL, the robot must remove a piece of meat from a grill and place it at a target location. Similar to PLACEBLOCK, we introduce two obstacles: one positioned between the gripper and the meat to constrain the approach trajectory, and another between the meat and the target region to constrain the carrying trajectory. This creates a combinatorial set of feasible strategies arising from different ways of approaching the meat and transporting it

to the target. The task is considered successful when the meat is fully removed from the grill and placed in the target region without contacting any obstacle; touching an obstacle at any stage causes the episode to fail.

- **OpenDoor.** Adapted from the robomimic benchmark (Mandlekar et al., 2021), OPENDOOR requires the robot to reach the handle of a door, turn the latch, and swing the door open to a target angle. Multimodality arises from the multiple feasible ways the end-effector can approach and engage the handle before pulling, producing distinct but equally valid opening trajectories under identical initial conditions. An episode is successful when the door is opened beyond the target angle within the horizon; failing to engage the handle or to open the door results in failure.

- **OpenMicrowave.** Built on the Franka Kitchen environment (Gupta et al., 2019), OPENMICROWAVE tasks a 9-DoF Franka arm with pulling open the microwave door in a cluttered kitchen scene. Because the handle can be reached and opened along several distinct approach paths among the surrounding kitchen fixtures, the task admits multiple valid execution modes. The episode is considered successful when the microwave door is opened past the target threshold.

- **Avoiding24 and Avoiding32.** Drawn from the D3IL benchmark (Jia et al., 2024), the AVOIDING tasks require the end-effector to travel from a fixed start location to a goal on the far side of a field of static obstacles. At each obstacle the agent may pass on either side, so the set of collision-free routes defines a large family of execution modes: **24** distinct modes in AVOIDING24 and **32** in AVOIDING32. An episode is successful if the end-effector reaches the goal without colliding with any obstacle; any collision terminates the episode as a failure.

**Multimodal Dataset Collection.** For each task, we explicitly construct a set of discrete execution modes to induce structured inter-mode diversity, and collect multiple noisy intra-mode demonstration variants for each mode to capture execution-level variability. Each mode corresponds to a distinct high-level strategy defined by geometric constraints in the environment (e.g., obstacle avoidance direction or grasping affordance), while demonstrations within the same mode differ due to stochasticity in execution and minor trajectory variations. This design yields datasets that exhibit both clearly separable inter-mode structure and realistic intra-mode noise, enabling a controlled evaluation of mode representation, selection, and adaptation. A summary of the number of modes, their geometric definitions, and the number of demonstrations collected per mode for each task is provided in Table 7.

| Task | # Modes | Demos / Mode | Mode Definition |
|---|---|---|---|
| CLOSEDRAWER | 8 | 10 | The gripper circumvents the obstacle placed between the gripper and drawer handle using one of four approach angles ($0°$, $90°$, $180°$, $270°$). For each angle, two distinct curvature profiles are used: one trajectory closer to the straight start–end line and one with a larger detour around the obstacle. |
| PICKUPCUP | 4 | 10 | Each mode corresponds to grasping the cup at one of four predefined grasping locations placed evenly around the rim at $0°$, $90°$, $180°$, and $270°$. The reaching trajectory is unconstrained, while the grasp location determines the mode. |
| PLACEBLOCK | 64 | 10 | The task is decomposed into two phases: reaching to grasp the block and carrying it to the target. Each phase follows the same mode construction as CLOSEDRAWER, with 8 modes defined by obstacle-circumventing approach angles and distances. The full task therefore admits $8 \times 8 = 64$ distinct modes from all combinations of reaching and carrying strategies. |
| MEATOFFGRILL | 64 | 10 | Similar to PLACEBLOCK, the task consists of a reaching phase (approaching and grasping the meat) and a carrying phase (transporting it to the target). Each phase has 8 modes defined by obstacle-avoidance angle and distance, yielding $8 \times 8 = 64$ total modes through combinatorial composition. |

*Table 7.* Multimodal dataset construction. For each task, we explicitly define discrete execution modes to induce inter-mode diversity, and collect multiple noisy demonstration variants within each mode to capture intra-mode variability.

### B.1.2. REAL ROBOT SETUP

In addition to simulation experiments, we evaluate PDP on a real-world manipulation task using a Franka Emika Panda robot arm. The task is OPENDRAWER, which requires the robot to reach a drawer handle, establish contact, and pull the drawer

open to a target configuration. The physical setup mirrors the simulation task at a high level but introduces substantially greater execution variability due to sensing noise, unmodeled dynamics, and human teleoperation.

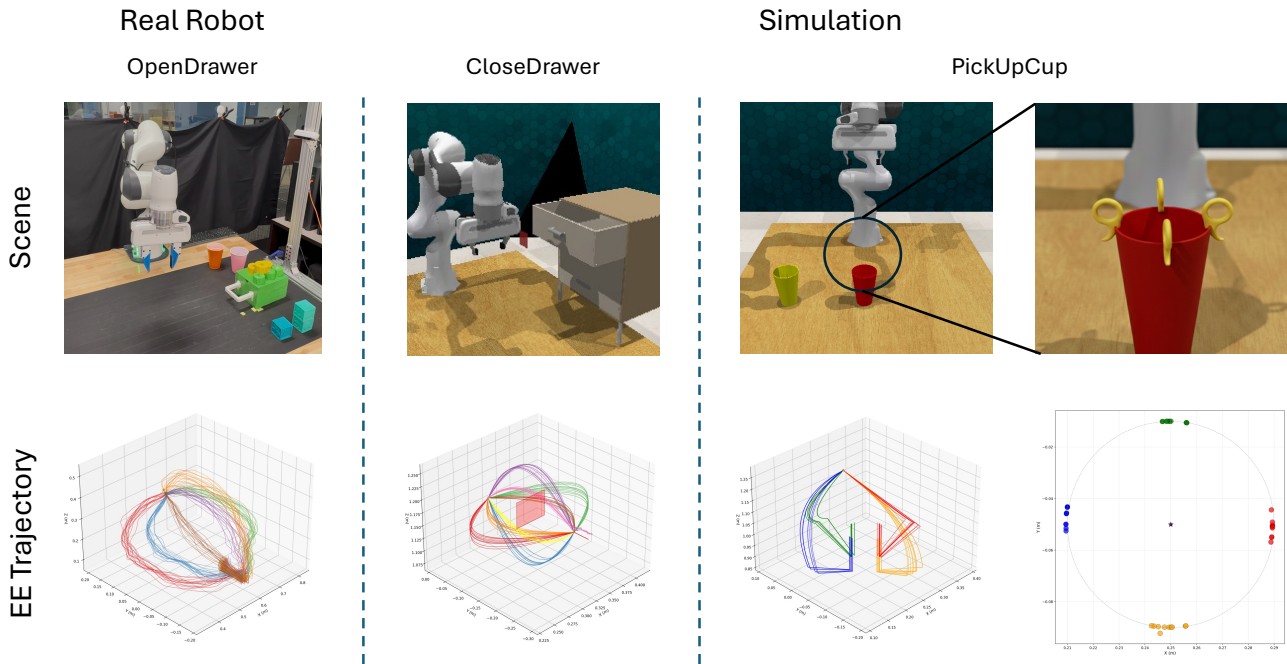

*Figure 7*. Visualization of multimodal demonstration datasets for three representative tasks. **First column:** Real-robot OPENDRAWER, showing the physical scene (top) and collected end-effector (EE) trajectories (bottom) from six distinct reaching-and-pulling modes. Trajectories of the same color correspond to noisy intra-mode demonstrations collected via SpaceMouse teleoperation. **Second column:** Simulated CLOSEDRAWER, where the bottom panel shows EE trajectories corresponding to different obstacle-circumventing reaching strategies around the drawer handle. **Third column:** Simulated PICKUPCUP, visualizing EE trajectories associated with different grasping strategies as the robot approaches the cup. **Fourth column:** A top-down visualization of grasping locations for PICKUPCUP, illustrating the four discrete grasping styles defined at angular positions $0°$, $90°$, $180°$, and $270°$ around the cup rim. Each grasping point corresponds to a distinct execution mode. Across all tasks, different colors denote distinct modes, while variations within a color reflect intra-mode execution noise.

To induce structured multimodality in the real-robot setting, we explicitly design six distinct execution modes for reaching and pulling the drawer handle. These modes differ in how the end-effector approaches the handle prior to contact. Specifically, the robot may approach from the left or right side of the handle, with two variants for each side corresponding to trajectories that remain closer to or farther from the straight start–end line. In addition, we include two vertical approach modes, where the end-effector reaches the handle from a higher or lower elevation relative to the handle center. Each mode therefore corresponds to a distinct geometric strategy for handle acquisition and pulling.

For each mode, we collect multiple noisy intra-mode demonstration trajectories via human teleoperation using a SpaceMouse. In total, we record 15 demonstrations per mode. Compared to simulation, these demonstrations exhibit substantially higher variability due to teleoperation imprecision and physical interaction effects, resulting in noticeably noisier end-effector trajectories even within the same mode. This increased intra-mode noise makes the real-robot dataset a particularly challenging testbed for behavior representation and mode conditioning.

### B.1.3. VISUALIZATION OF TRUE MULTIMODAL DATASET COLLECTION

Figure 7 depicts representative multimodal demonstration data from three tasks: real-robot OPENDRAWER, and simulated CLOSEDRAWER and PICKUPCUP. For each task, we show the task scene (top row) together with the corresponding end-effector (EE) trajectories from the collected demonstrations (bottom row). Different colors indicate distinct execution modes, while trajectories within the same color correspond to noisy intra-mode variants. These visualizations highlight the structured inter-mode diversity induced by our task design, as well as the substantial intra-mode variability present in the demonstrations. In particular, the real-robot OPENDRAWER trajectories exhibit significantly higher noise and dispersion

compared to simulation, reflecting teleoperation imprecision and real-world contact dynamics. Together, these examples illustrate that our datasets capture true multimodality beyond simple stochastic perturbations, providing a challenging benchmark for behavior representation, mode selection, and adaptation.

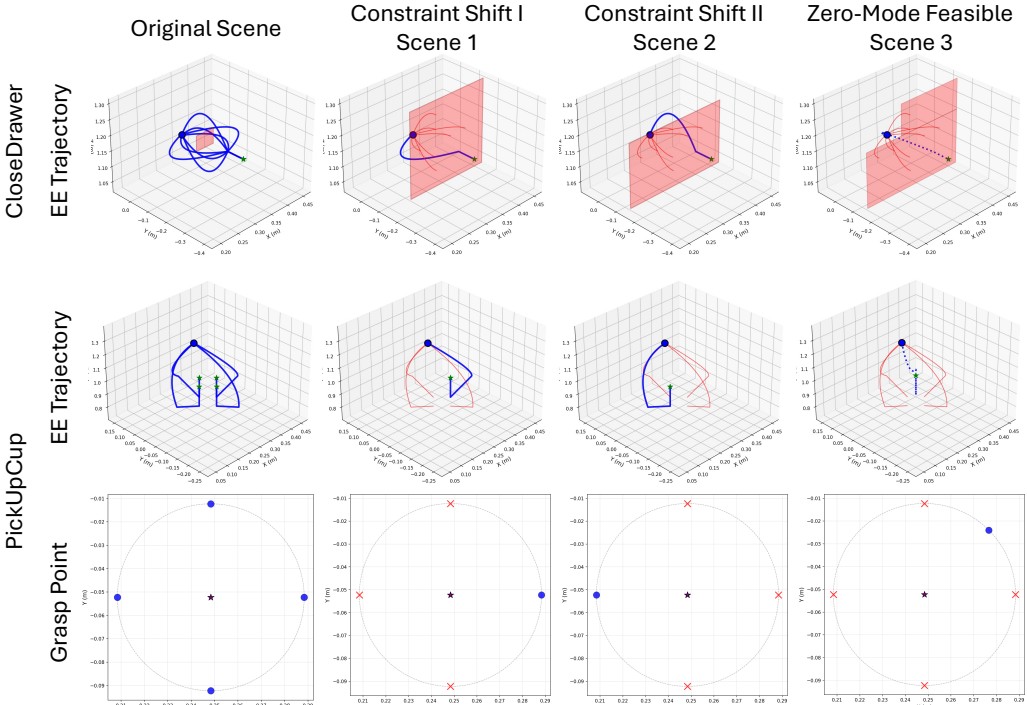

*Figure 8.* Examples of constraint-induced behavior shifts for two representative tasks under four evaluation variants. **Top row:** CLOSEDRAWER, showing EE trajectories for the original training scene and three constraint-shifted scenes. **Middle row:** PICKUPCUP EE trajectories, illustrating different approach behaviors toward the cup under the same four scene variants. **Bottom row:** PICKUPCUP grasp-point visualizations, showing the grasping locations on the cup rim that define the four training modes. Columns correspond to (from left to right): *Original Scene*, *Constraint Shift I (Scene 1)*, *Constraint Shift II (Scene 2)*, and *Zero-Mode Feasible (Scene 3)*. Solid trajectories indicate execution modes covered by the training dataset (eight for CLOSEDRAWER, four for PICKUPCUP). Red trajectories indicate failure due to collision with obstacles or invalid grasping, causing early termination. The dotted trajectory denotes a newly provided demonstration that is not present in the training data and is required to solve the task in the zero-mode-feasible setting.

## B.2. Additional Results

Due to space constraints, the main text (Table 2) reports constraint-shifted results on four representative domains. Table 8 provides the complete results across all eight manipulation domains and all nine methods, under the same evaluation protocol described in Appendix B.3.

Table 8 reports the complete constraint-shifted results across all eight domains and nine methods. PDP attains the highest overall average success rate ($95.0\%$), exceeding the strongest baseline by more than $50$ points, and is the only method that remains robust across every domain and every scene type. Consistent with the main text, several baselines are competitive on individual domains—VQ-BeT on CLOSEDRAWER, BC-GMM on OPENDOOR, and Diff-ES on the AVOIDING tasks—but none generalizes across the full benchmark. In particular, every baseline degrades sharply in the zero-mode-feasible setting (Scene 3), where success requires synthesizing a behavior absent from the training data, whereas PDP sustains high success by fitting its geometry-aligned latent to the single provided demonstration. These full results reinforce the main-text conclusion that an explicit, behavior-aligned latent offers a more reliable adaptation interface.

## B.3. Evaluation Variants and Constraint Shifts

We evaluate PDP and all baseline methods under a set of controlled environment variants designed to isolate behavior-side adaptation under constraint-induced shifts. Each task is tested under four conditions: the *Original Scene*, which matches the training environment and serves as a baseline for multimodal imitation fidelity; *Scene 1* and *Scene 2*, which introduce

*Table 8.* Full success rate (%) on **constraint-shifted** scenes across all eight manipulation domains and nine methods (mean±std over 3 seeds). Scene 1,2 invalidate all except a single training mode; Scene 3 is the **zero-mode-feasible** setting, where all training modes are invalidated and a single new demonstration is provided for adaptation. The best performance in each row is shown in bold. This table extends the four-domain summary in Table 2.

| Task | Scene | PDP | DP | BC-GMM | BC | IBC | ADPro | Diff-ES | VQ-BeT | BESO |
|---|---|---|---|---|---|---|---|---|---|---|
| CLOSEDRAWER | 1 | **100.0 ± 0.0** | **100.0 ± 0.0** | 0.0 ± 0.0 | 0.0 ± 0.0 | 0.0 ± 0.0 | **100.0 ± 0.0** | 99.2 ± 0.7 | **100.0 ± 0.0** | 92.5 ± 2.5 |
| | 2 | **100.0 ± 0.0** | 88.3 ± 14.2 | 0.0 ± 0.0 | 82.5 ± 4.3 | 0.0 ± 0.0 | 96.7 ± 1.8 | 34.2 ± 2.5 | **100.0 ± 0.0** | 97.5 ± 2.5 |
| | 3 | **100.0 ± 0.0** | 40.0 ± 13.0 | 0.0 ± 0.0 | 2.5 ± 2.5 | 0.0 ± 0.0 | 0.0 ± 0.0 | 1.7 ± 0.7 | **100.0 ± 0.0** | **100.0 ± 0.0** |
| PLACEBLOCK | 1 | **95.0 ± 2.5** | 0.0 ± 0.0 | 0.0 ± 0.0 | 12.5 ± 3.8 | 0.0 ± 0.0 | 0.0 ± 0.0 | 14.2 ± 4.8 | 90.0 ± 2.5 | 0.0 ± 0.0 |
| | 2 | **95.8 ± 1.4** | 0.0 ± 0.0 | 0.0 ± 0.0 | 2.5 ± 2.5 | 0.0 ± 0.0 | 0.0 ± 0.0 | 13.3 ± 5.4 | 92.2 ± 0.9 | 22.5 ± 15.9 |
| | 3 | **86.7 ± 10.1** | 2.5 ± 2.5 | 0.0 ± 0.0 | 5.0 ± 2.5 | 0.0 ± 0.0 | 0.0 ± 0.0 | 13.3 ± 5.4 | 76.0 ± 3.1 | 16.2 ± 11.5 |
| MEATOFFGRILL | 1 | **84.2 ± 1.4** | 0.0 ± 0.0 | 0.0 ± 0.0 | 80.0 ± 5.0 | 0.0 ± 0.0 | 0.0 ± 0.0 | 0.0 ± 0.0 | 0.0 ± 0.0 | 0.0 ± 0.0 |
| | 2 | **86.7 ± 5.2** | 0.0 ± 0.0 | 0.0 ± 0.0 | 65.0 ± 3.8 | 0.0 ± 0.0 | 0.8 ± 0.7 | 0.0 ± 0.0 | 7.5 ± 2.5 | 0.0 ± 0.0 |
| | 3 | **91.7 ± 2.9** | 2.5 ± 2.5 | 0.0 ± 0.0 | 0.0 ± 0.0 | 0.0 ± 0.0 | 0.8 ± 0.7 | 2.5 ± 2.5 | 0.0 ± 0.0 | 0.0 ± 0.0 |
| PICKUPCUP | 1 | **95.8 ± 1.4** | 7.5 ± 13.0 | 92.5 ± 2.5 | 2.5 ± 2.5 | 0.0 ± 0.0 | 0.0 ± 0.0 | 20.0 ± 4.1 | 85.0 ± 2.5 | 0.0 ± 0.0 |
| | 2 | **85.0 ± 2.5** | 0.8 ± 1.4 | 72.5 ± 4.3 | 4.2 ± 2.9 | 0.0 ± 0.0 | 10.8 ± 4.9 | 20.0 ± 3.5 | 55.0 ± 5.0 | 22.5 ± 15.9 |
| | 3 | **82.5 ± 4.3** | 15.8 ± 21.0 | 0.0 ± 0.0 | 22.5 ± 4.3 | 0.0 ± 0.0 | 3.3 ± 1.4 | 20.0 ± 2.5 | 70.0 ± 2.5 | 16.2 ± 11.5 |
| OPENDOOR | 1 | 98.3 ± 1.7 | 22.5 ± 3.8 | **100.0 ± 0.0** | 0.0 ± 0.0 | 0.0 ± 0.0 | 25.0 ± 2.9 | 92.0 ± 3.5 | 0.0 ± 0.0 | 0.0 ± 0.0 |
| | 2 | 96.7 ± 1.7 | 39.2 ± 5.8 | **100.0 ± 0.0** | 78.0 ± 4.1 | 0.0 ± 0.0 | 51.7 ± 2.2 | 90.0 ± 2.5 | 0.0 ± 0.0 | 0.0 ± 0.0 |
| | 3 | **87.5 ± 12.3** | 19.2 ± 0.8 | 0.0 ± 0.0 | 0.0 ± 0.0 | 0.0 ± 0.0 | 16.7 ± 0.8 | 62.0 ± 5.7 | 0.0 ± 0.0 | 0.0 ± 0.0 |
| OPENMICROWAVE | 1 | **98.3 ± 1.2** | 12.5 ± 1.4 | 0.0 ± 0.0 | 6.7 ± 2.4 | 0.0 ± 0.0 | 17.5 ± 2.0 | 16.7 ± 4.2 | 61.7 ± 3.1 | 35.0 ± 10.2 |
| | 2 | **100.0 ± 0.0** | 45.0 ± 12.7 | 0.0 ± 0.0 | 18.3 ± 2.4 | 0.0 ± 0.0 | 61.7 ± 3.1 | 46.7 ± 5.1 | **100.0 ± 0.0** | 60.8 ± 8.5 |
| | 3 | **100.0 ± 0.0** | 27.5 ± 3.5 | 0.0 ± 0.0 | 1.7 ± 1.2 | 0.0 ± 0.0 | 45.0 ± 2.0 | 36.7 ± 6.6 | 96.7 ± 2.4 | 50.0 ± 7.4 |
| AVOIDING24 | 1 | **100.0 ± 0.0** | 13.8 ± 2.4 | 0.0 ± 0.0 | 33.3 ± 4.7 | 0.0 ± 0.0 | 11.2 ± 5.5 | **100.0 ± 0.0** | 0.0 ± 0.0 | 0.0 ± 0.0 |
| | 2 | **100.0 ± 0.0** | 8.8 ± 1.4 | 0.0 ± 0.0 | 33.3 ± 2.4 | 0.0 ± 0.0 | 5.0 ± 1.4 | **100.0 ± 0.0** | 0.0 ± 0.0 | 0.0 ± 0.0 |
| | 3 | **100.0 ± 0.0** | 0.0 ± 0.0 | 0.0 ± 0.0 | 33.3 ± 5.9 | 0.0 ± 0.0 | 0.8 ± 1.2 | 39.2 ± 2.4 | 0.0 ± 0.0 | 0.0 ± 0.0 |
| AVOIDING32 | 1 | **100.0 ± 0.0** | 16.2 ± 3.1 | 0.0 ± 0.0 | 66.7 ± 3.1 | 0.0 ± 0.0 | 11.2 ± 4.7 | **100.0 ± 0.0** | 0.0 ± 0.0 | 0.0 ± 0.0 |
| | 2 | **100.0 ± 0.0** | 13.8 ± 2.4 | 0.0 ± 0.0 | 66.7 ± 4.7 | 0.0 ± 0.0 | 15.0 ± 3.5 | **100.0 ± 0.0** | 0.0 ± 0.0 | 0.0 ± 0.0 |
| | 3 | **95.8 ± 1.2** | 0.0 ± 0.0 | 0.0 ± 0.0 | 66.7 ± 2.4 | 0.0 ± 0.0 | 0.0 ± 0.0 | 0.0 ± 0.0 | 0.0 ± 0.0 | 0.0 ± 0.0 |
| *Overall Avg.* | | **95.0** | 19.8 | 15.2 | 28.5 | 0.0 | 19.7 | 42.6 | 43.1 | 21.4 |

constraints that invalidate a subset of the demonstrated strategies while leaving exactly one training mode feasible; and *Scene 3*, a zero-mode-feasible setting in which all training modes fail and success requires discovering a qualitatively new trajectory guided by a single new demonstration.

Figure 8 illustrates these variants using two representative tasks: CLOSEDRAWER and PICKUPCUP. Solid trajectories denote execution modes covered by the training dataset (eight for CLOSEDRAWER and four for PICKUPCUP). In constraint-shifted scenes, trajectories shown in red indicate failure due to collision with obstacles or invalid grasping, causing the episode to terminate early. The dotted trajectory corresponds to a newly provided demonstration that lies outside the training distribution and is required to solve the task in the zero-mode-feasible setting. We include these examples for illustration; the remaining tasks follow analogous constraint constructions. For the two pick-and-place tasks (PLACEBLOCK and MEATOFFGRILL), walls are inserted both between the initial gripper position and the object and between the object and the target, yielding constraint patterns similar to CLOSEDRAWER. For the real-robot OPENDRAWER task, physical objects such as cups, books, and snack containers are placed between the robot and the drawer to create realistic constraint-induced behavior shifts, as shown in Figure 9.

## B.4. Policy Training and Baseline Configurations

We evaluate Parameterized Diffusion Policy (PDP) and a set of representative behavior cloning baselines under a unified training and evaluation protocol. All methods share the same backbone architecture and observation processing pipeline to ensure a fair comparison; differences arise only from their learning objectives, conditioning mechanisms, and test-time adaptation strategies.

PDP is trained using the joint optimization scheme described in Algorithm 2, alternating between embedding refinement via the geometry-aligned objective $\mathcal{L}_{embed}$ and diffusion denoiser optimization via $\mathcal{L}_{DP}$. The denoiser is conditioned on the behavior latent $z$ through global modulation, enabling deep and explicit integration of the control signal. At evaluation time in the original (non-shifted) environments, we compute a representative latent for each mode by averaging the latent codes of demonstrations belonging to that mode, and execute the policy conditioned on this mean latent. For each task, we

| Original Scene | Constraint Shift I Scene 1 | Constraint Shift II Scene 2 | Zero-Mode Feasible Scene 3 |
|---|---|---|---|

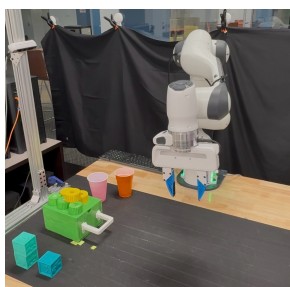 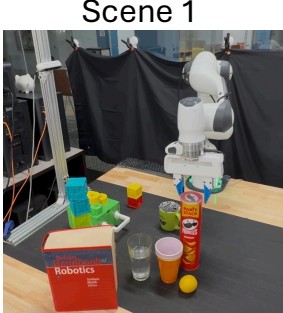 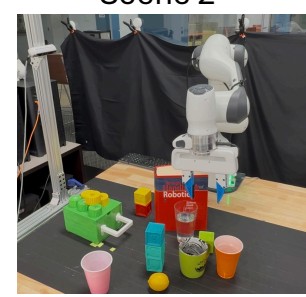 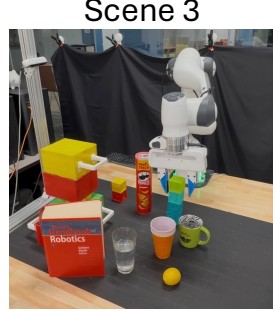

*Figure 9.* Real-robot constraint-induced behavior shift for OPENDRAWER. To simulate realistic deployment conditions, everyday objects such as cups, books, and snack containers are placed between the robot and the drawer, blocking previously demonstrated reaching strategies. These physical constraints invalidate training modes and require the policy to adapt its approach trajectory under real-world noise and contact dynamics.

| Method | Training Objective | Test-Time Adaptation | Evaluation Protocol |
|---|---|---|---|
| PDP | Diffusion loss $\mathcal{L}_{DP}$ with geometry-aligned latent learning $\mathcal{L}_{embed}$ | Optimize or select behavior latent $z$; denoise with frozen weights | 40 rollouts total; original scene uses per-mode mean $z$; constraint-shifted scenes fit $z$ from a single demo |
| DP | Diffusion noise prediction loss | Optimize denoising noise seed at test time (noise-space adaptation) | 40 rollouts with stochastic sampling or noise optimization |
| BC | Mean-squared error on actions | Fine-tune policy weights with small learning rate | 40 rollouts after fine-tuning on new demonstration |
| BC-GMM | Negative log-likelihood under mixture model | Select mixture component via posterior likelihood | 40 rollouts using MAP mixture selection |
| IBC | Energy-based imitation loss | Bias inference toward target demonstration | 40 rollouts with energy-guided inference |

*Table 9.* Comparison of training objectives, test-time adaptation mechanisms, and evaluation protocols across PDP and baseline methods. All methods use the same backbone architecture and observation encoder for fair comparison.

perform a total of 40 rollouts, distributed evenly across modes.

For long-horizon tasks (PLACEBLOCK and MEATOFFGRILL), demonstrations are segmented into reaching and carrying phases based on gripper open/close events. Each phase is normalized by its start and end positions prior to embedding, so that the behavior latent captures trajectory shape rather than absolute location. During evaluation, the same latent $z$ is used to condition both phases, resulting in eight effective latents for these tasks despite their two-stage structure.

We compare PDP against four baseline paradigms: standard Diffusion Policy (DP), vanilla Behavior Cloning (BC), Behavior Cloning with Gaussian Mixture Models (BC-GMM), and Implicit Behavioral Cloning (IBC). All baselines are trained on the same datasets and evaluated using 40 rollouts per task in both original and constraint-shifted environments. Baseline-specific adaptation and inference procedures follow standard practice and are summarized in Table 9.

# C. Extended Experimental Results

## C.1. Training Dynamics of Latent Integration Mechanisms

To better understand why deep latent integration improves adaptation performance, we analyze the diffusion training dynamics under different latent integration mechanisms. Figure 10 plots the diffusion noise-prediction loss throughout training for three variants: (i) *Global Modulation*, which integrates the behavior latent via affine modulation of intermediate feature maps, (ii) *Concatenation*, which appends the latent $z$ to the denoiser input, and (iii) *Unconditioned*, a standard diffusion policy without a behavior latent.

| Hyperparameter | Value |
| --- | --- |
| Diffusion steps $K$ | 20 |
| Behavior latent dimension $d_z$ | 2 |
| Embedding learning rate | $1 \times 10^{-4}$ |
| Denoiser learning rate | $1 \times 10^{-4}$ |
| Batch size | 256 |
| Global Modulation hidden dimension | [16, 32] |
| $\beta_{\text{KL}}$ (VAE) | $1 \times 10^{-3}$ |
| $\beta_{\text{geo}}$ (geometry loss) | 1.0 |
| Soft-DTW smoothing $\gamma$ | $1 \times 10^{-5}$ |

*Table 10.* Training hyperparameters used for PDP and baseline methods. Unless otherwise specified, the same architectural and optimization settings are shared across methods.

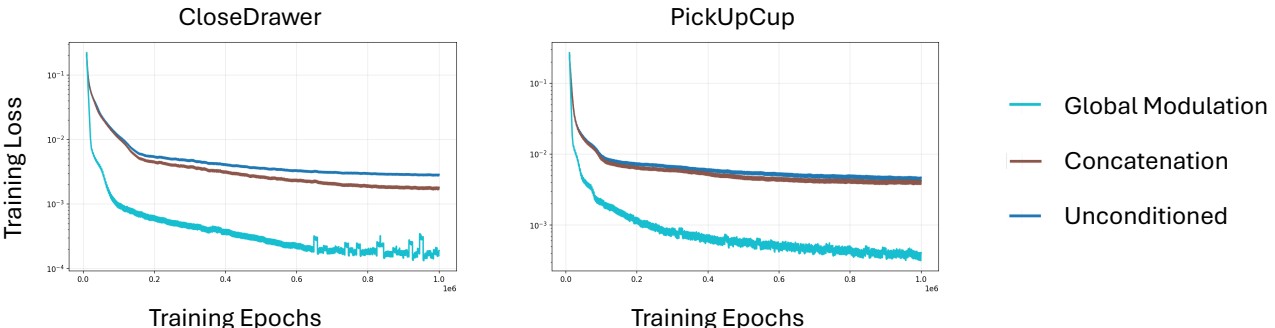

*Figure 10.* Diffusion training loss under different latent integration mechanisms. Left: CLOSEDRAWER. Right: PICKUPCUP. Global Modulation consistently converges to a substantially lower loss than Concatenation and Unconditioned variants, indicating reduced inter-mode ambiguity during training.

Across both CLOSEDRAWER and PICKUPCUP, Global Modulation consistently achieves a substantially lower training loss—nearly an order of magnitude lower at convergence—than the other two variants. In contrast, Concatenation and Unconditioned models converge to similar loss levels, indicating that shallow latent injection provides limited benefit for resolving mode ambiguity during training.

This behavior directly supports our analysis in Appendix A.4. When the denoiser is weakly or not conditioned on behavior, it must implicitly average over multiple incompatible trajectory modes, resulting in a harder regression problem with higher irreducible variance. Global Modulation collapses this inter-mode ambiguity by explicitly reparameterizing the denoiser around a target behavior, converting diffusion from a global mixture-modeling task into a mode-specific denoising problem. The resulting reduction in training loss reflects a genuine simplification of the learning objective rather than improved optimization alone, and explains why Global Modulation yields more stable and effective test-time adaptation.

### C.2. Comparative Trajectory Visualizations

Quantitative success rates summarize whether a rollout terminates in success, but they often under-specify *how* that success is achieved and can obscure qualitatively different failure modes. This is particularly important in our setting of *constraint-induced behavior shifts*, where feasibility hinges on selecting (or synthesizing) a specific geometric strategy: two methods may have the same binary outcome on a subset of trials while exhibiting radically different levels of consistency, safety margin, and mode fidelity. We therefore complement Figure 11—12 with trajectory-level visualizations that directly reveal (i) whether a policy executes a coherent mode versus averaging across incompatible modes, (ii) whether the induced motion remains within the feasible corridor under new constraints.

**Simulation: CLOSEDRAWER (row 1 in Fig. 11).** The CLOSEDRAWER scene is multimodal because the end-effector must route around an obstacle to reach the handle, yielding multiple distinct approach geometries. In Fig. 11 (top row), PDP produces smooth, mode-consistent trajectories that remain concentrated around a small set of feasible approach manifolds (multiple colored rollouts indicate consistent reconstruction/selection of distinct modes). In contrast, unconditioned DP

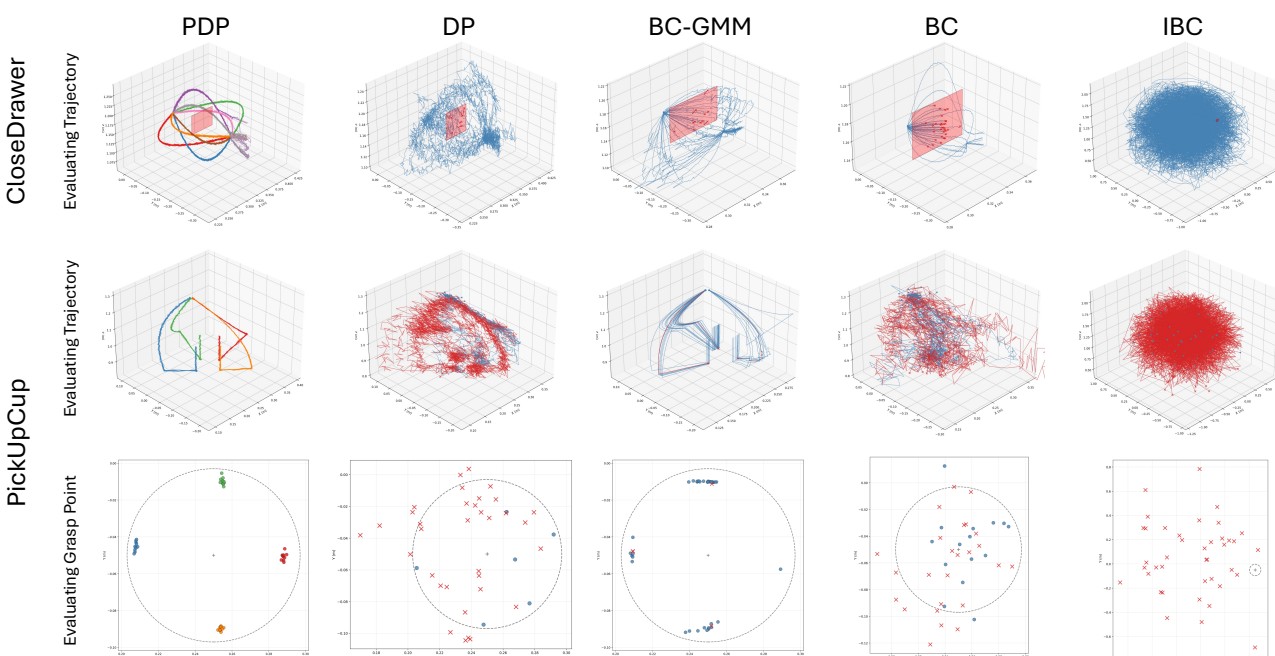

*Figure 11.* **Simulation trajectory visualizations under constraint-induced shifts (CloseDrawer, PickUpCup).** Columns compare PDP against DP, BC-GMM, BC, and IBC. *Top row (CloseDrawer):* executed end-effector trajectories; PDP remains mode-consistent and concentrated along feasible obstacle-circumventing corridors, while unconditioned baselines exhibit mode interference and collapse into infeasible regions. *Middle row (PickUpCup):* executed end-effector trajectories; *Bottom row (PickUpCup):* realized grasp points on the cup rim. PDP commits to valid grasp affordances with low dispersion, whereas unconditioned baselines produce scattered (often invalid) grasp attempts, reflecting cross-mode averaging and unstable steering.

exhibits pronounced *mode interference*: repeated rollouts spread widely and frequently drift into the obstacle region (visualized by the dense, noisy trajectory bundle that "fills" the workspace rather than tracking a narrow approach corridor). This is the qualitative signature of the ambiguity described in Appendix A.4: when the denoiser is not explicitly indexed by behavior, it must implicitly average denoising directions across incompatible strategies, producing unstable rollouts that collapse into infeasible regions. BC-GMM can partially preserve a small number of coarse modes, but the resulting trajectories still show substantial dispersion near the constraint boundary; BC and IBC further degenerate into highly scattered behavior, reflecting the lack of a reliable mechanism to *select* a single geometric strategy under ambiguity.

**Simulation: PICKUPCUP (rows 2–3 in Fig. 11).** PICKUPCUP isolates a different kind of multimodality: the dominant discrete choice is the *grasp affordance* (four valid grasp points around the rim), while approach motions are conditioned on that choice. Fig. 11 makes this distinction explicit by pairing trajectory rollouts (middle row) with the realized grasp points (bottom row). PDP concentrates its grasp selections tightly around the valid affordances, indicating that the learned latent provides a stable control handle for committing to a single grasp mode and executing it consistently. Unconditioned DP, despite its expressive generative capacity, shows "averaging" behavior in grasp selection: realized grasp points disperse broadly along the rim and into invalid regions, matching the noisy and inconsistent approach trajectories. BC-GMM can sometimes select a plausible component but remains brittle under shifts; BC and IBC scatter across many invalid grasp attempts, indicating that small errors in action prediction translate into qualitatively wrong grasp commitment. The key takeaway is that the advantage of PDP here is not merely smoother motion, but *mode commitment*: the latent-conditioned denoiser resolves the discrete ambiguity early and then performs within-mode denoising, preventing cross-mode averaging from manifesting as invalid grasp selection.

**Real robot: OPENDRAWER (Fig. 12).** On hardware, execution noise, contact variability, and perception imperfections amplify the cost of ill-conditioned steering: a method that only sporadically finds a correct strategy may still be unusable if its rollouts jitter across modes or repeatedly skim constraints. Fig. 12 shows that PDP maintains coherent, repeatable approach-and-pull trajectories across trials, despite real-world stochasticity, whereas DP exhibits substantially higher dispersion and inconsistent approach geometry. This qualitative difference matches the real-robot success-rate gap (Table 3): PDP's

behavior latent acts as a low-dimensional, geometry-aligned interface that stabilizes test-time steering, while noise-space control in DP remains sensitive and produces high-variance rollouts under constraint changes.

## Real Robot OpenDrawer

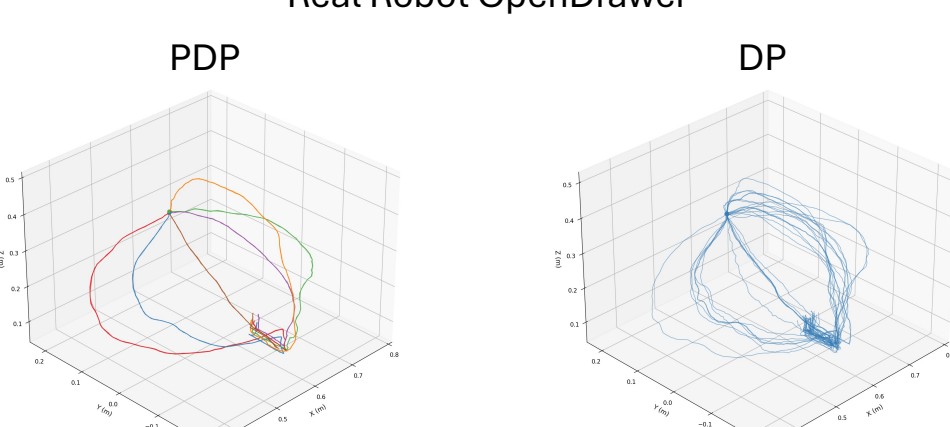

*Figure 12.* **Real-robot OpenDrawer trajectory visualizations under constraint-induced shifts.** PDP produces repeatable, coherent approach-and-pull trajectories across trials despite hardware noise, while DP exhibits substantially larger dispersion and inconsistent approach geometry, illustrating the instability of noise-space steering under real-world constraints.

**Overall takeaway.** Across simulation and hardware, these visualizations corroborate the central claim of the paper: PDP converts diffusion from noise-driven diversity into a controllable mechanism by exposing a geometry-aligned behavior coordinate. Trajectory plots reveal *how* this translates into practice—stable mode selection, reduced cross-mode averaging, and safer constraint-aware execution—properties that are difficult to diagnose from binary success rates alone.

### C.3. Latent Space Navigation and Interpolation

In this section, we analyze how the learned behavior manifold supports controlled interpolation across qualitatively different task semantics (Fig. 13–15). These plots are designed to probe not whether PDP can reconstruct training modes, but whether the geometry induced by $\mathcal{L}_{\text{geo}}$ yields a *navigable* space in which intermediate latents correspond to coherent, physically meaningful behaviors.

In CLOSEDRAWER (Fig. 13), the latent space encodes reaching styles around an obstacle. We evaluate multiple interpolation paths—including linear and curved traversals—between distant mode clusters. Despite traversing different trajectories in latent space, all interpolations produce smooth, collision-free end-effector paths that continuously deform the approach geometry. This demonstrates that PDP does not merely interpolate between memorized trajectories, but instead induces a continuous family of feasible reaching strategies parameterized by $z$.

In contrast, PICKUPCUP exhibits a different semantic structure: the primary source of multimodality lies in discrete grasp affordances rather than approach paths. As shown in Fig. 14, interpolating $z$ produces a smooth progression of grasp locations along the rim of the cup. This confirms that the latent space does not collapse all tasks into a single notion of trajectory similarity; instead, it adapts to the task-specific factors of variation encoded by the demonstrations.

Finally, Fig. 15 replicates the same interpolation procedure on a real robot, where demonstrations exhibit substantial execution noise and unmodeled dynamics. Despite this, interpolated latents yield consistent qualitative changes in end-effector trajectories, indicating that the learned manifold remains coherent beyond simulation. Together, these results support our claim that PDP learns a geometry-aligned behavior space in which interpolation corresponds to semantically meaningful generalization, rather than arbitrary mixtures of training behaviors.

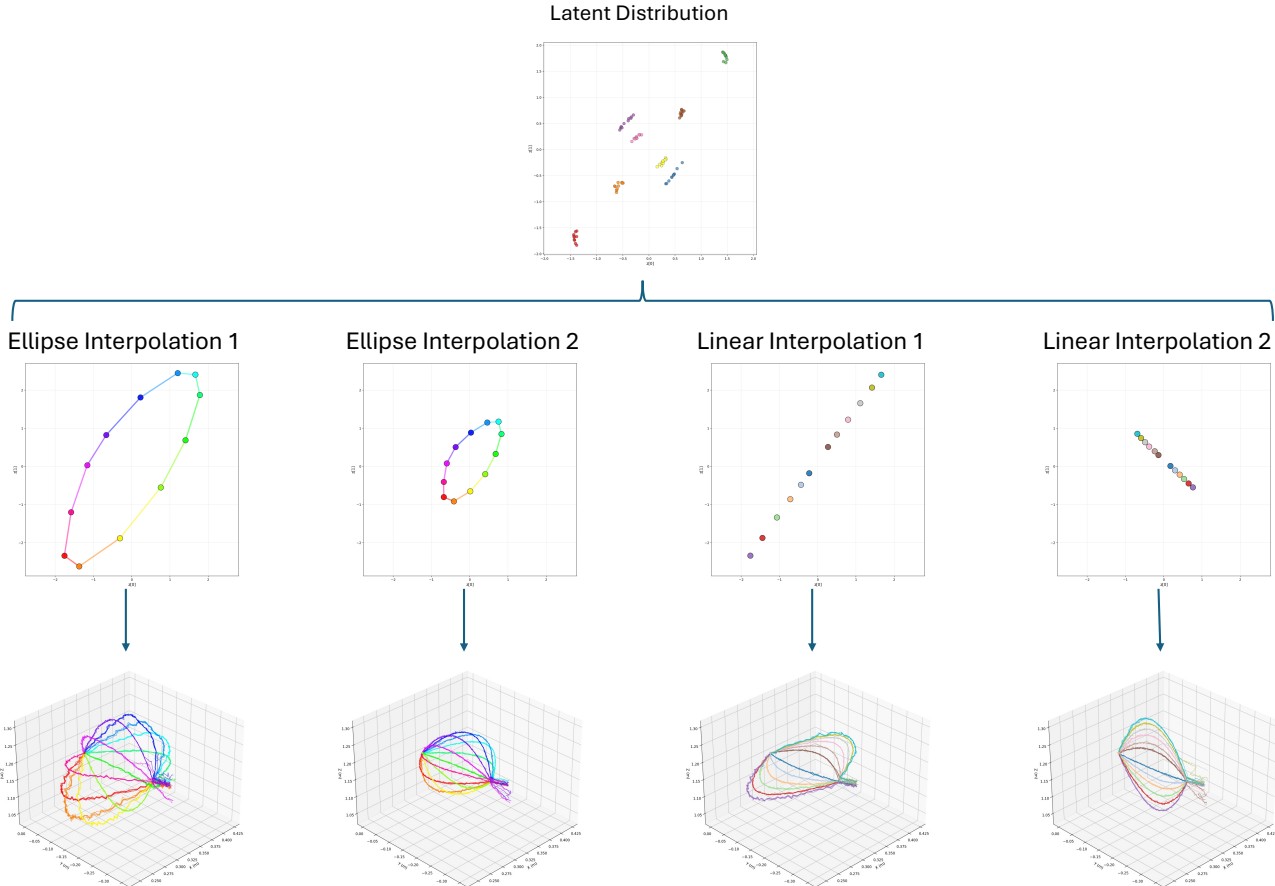

*Figure 13.* **Latent space interpolation on CLOSEDRAWER (simulation).** Top: learned latent distribution with clusters corresponding to distinct reaching strategies. Middle: multiple interpolation paths in latent space, including linear and elliptical traversals between clusters. Bottom: executed end-effector trajectories produced by conditioning PDP on interpolated latents. Despite differing interpolation geometries in $z$, the resulting behaviors exhibit smooth, physically plausible transitions in approach style and obstacle avoidance, confirming that the latent manifold is continuous and navigable.

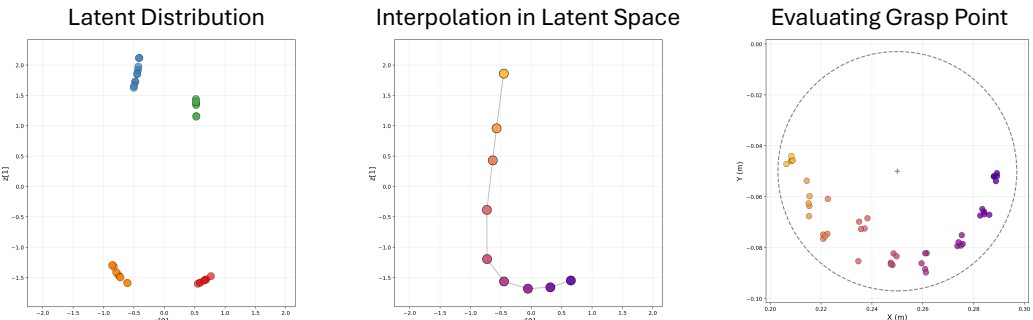

*Figure 14.* **Latent interpolation for grasp selection on PICKUPCUP.** Left: latent distribution with clusters corresponding to discrete grasp affordances. Middle: interpolation between grasp-related latents. Right: evaluated grasp points on the cup rim. Unlike reaching-dominated tasks, interpolation here induces a smooth shift in grasp location along the cup edge, demonstrating that the latent space captures task-specific semantics beyond trajectory shape alone.

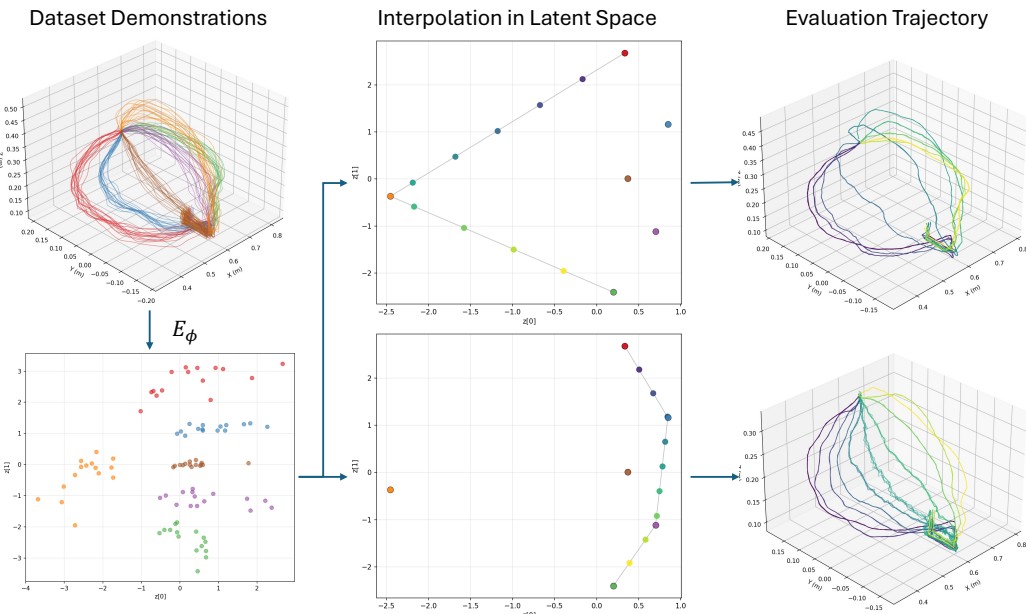

*Figure 15.* **Latent space interpolation on a real robot.** Left: demonstration trajectories collected on hardware. Middle: interpolated latents in the learned behavior space. Right: executed end-effector trajectories under latent interpolation. Despite substantial execution noise and unmodeled dynamics, interpolated latents yield consistent and smoothly varying behaviors, indicating that the learned manifold generalizes beyond simulation.

