# OpenReview forum: "From Noise to Control: Parameterized Diffusion Policies"
_ICML.cc/2026/Conference — ICML 2026 regular_

### Official Review · Reviewer_wwsZ · 2026-02-24

**Soundness:** 2
**Presentation:** 2
**Significance:** 3
**Originality:** 3
**Overall Recommendation:** 4
**Confidence:** 4

**Summary:**

This paper introduces a novel approach called PDP to address the challenge of constraint-induced behavior shifts in AI systems, where new environmental obstacles invalidate previously learned trajectories. Unlike standard Diffusion Policy (DP), which suffers from highly unstable and unpredictable trajectory changes when steered via sampling noise, PDP leverages a structured behavior latent variable $z$. Because the geometry of this latent space is inherently aligned with trajectory similarity, small adjustments to $z$ result in smooth, predictable, and stable changes in the model's actions. Ultimately, this mechanism enables continuous interpolation and robust adaptation, allowing the AI to safely navigate and overcome new physical constraints without erratic movements.

**Compliance With Llm Reviewing Policy:**

Affirmed.

**Final Justification:**

This work has a very interesting starting point, proposing a PDP control method. It can effectively solve the problem of control trajectory generalization. The authors also added experimental supplements in their response, so I chose to raise the score.

**Key Questions For Authors:**

See weaknesses.

**Limitations:**

Please add "limitation section" to the main part.

**Strengths And Weaknesses:**

## Strengths
- Tackles realistic physical constraints: Shifts focus from simple visual changes to complex physical obstacles that force the AI to find new movement paths.
- Fixes a critical flaw in standard DP: Points out that steering standard Diffusion Policy via noise is unstable, where tiny tweaks cause unpredictable, wild movements.
- Introduces a precise control mechanism: Proposes PDP, which uses a structured latent variable ($z$) that perfectly aligns with how similar different trajectories are.

## Weaknesses
1. The empirical evaluation is limited to a small suite of manipulation domains. To substantiate the claim of general controllable diffusion policies, experiments on standardized benchmarks such as D4RL locomotion, Franka Kitchen, and Meta-World would significantly strengthen the paper.

2. The experimental setup favors PDP by providing a successful demonstration at test time for latent fitting. In contrast, gradient- or reward-guided methods typically adapt without access to such demonstrations. This makes the comparison potentially inequitable and may confound the source of performance gains.

3. While the empirical results are promising, the experimental section lacks comparison against stronger test-time adaptation baselines, particularly recent manifold-constrained denoising methods (e.g., ADPro[1]) , advanced noise-space optimization strategies and reward-guided methods(e.g., Diffusion-ES[2]). As a result, it is difficult to assess whether the gains stem from the proposed geometry-aligned latent manifold, or from weaker implementations of existing adaptation techniques.

4. The paper lacks a rigorous theoretical analysis of the proposed geometry alignment. While a proportional relationship between latent distance and trajectory distance is enforced empirically, it remains unclear how precise this correspondence is. In particular, is there a formal bound on the distortion between the two metrics, or any guarantee on approximation error? Providing such analysis would strengthen the theoretical foundation of the method.

5. The training pipeline is relatively complex, and it is unclear how robust it is in practice. In particular, compressing high-dimensional trajectories into a low-dimensional latent space may lead to information loss. How is the latent dimensionality selected, and is there any sensitivity analysis? Moreover, while a VAE objective is used, it does not inherently guarantee the avoidance of mode collapse. Is there theoretical or empirical evidence showing that the learned latent manifold preserves multimodal structure without collapsing distinct behaviors?

6. The related work section could be strengthened by incorporating more recent diffusion-based control and RL-guided methods, such as DIPO[3], QSM[4], and DACER[5][6].

**The paper presents very interesting ideas, and I would be willing to raise my grade if the aforementioned issues can be resolved.**

[1] Li Z, Yang R, Chen R, et al. ADPro: a Test-time Adaptive Diffusion Policy via Manifold-constrained Denoising and Task-aware Initialization for Robotic Manipulation[J]. arXiv preprint arXiv:2508.06266, 2025.

[2] Yang B, Su H, Gkanatsios N, et al. Diffusion-es: Gradient-free planning with diffusion for autonomous and instruction-guided driving[C]//Proceedings of the IEEE/CVF conference on computer vision and pattern recognition. 2024: 15342-15353.

[3] Yang L, Huang Z, Lei F, et al. Policy representation via diffusion probability model for reinforcement learning[J]. arXiv preprint arXiv:2305.13122, 2023.

[4] Psenka M, Escontrela A, Abbeel P, et al. Learning a diffusion model policy from rewards via q-score matching[J]. arXiv preprint arXiv:2312.11752, 2023.

[5] Wang Y, Wang L, Jiang Y, et al. Diffusion actor-critic with entropy regulator[J]. Advances in Neural Information Processing Systems, 2024, 37: 54183-54204.

[6] Wang Y, Wang L, Tan M, et al. Enhanced dacer algorithm with high diffusion efficiency[J]. arXiv preprint arXiv:2505.23426, 2025.

---

> ### Author Rebuttal · Authors · 2026-03-31
>
> We thank the reviewer for their positive assessment, for highlighting our method’s significance, and for noting that it solves a critical flaw in standard Diffusion Policies. Below, we address their questions.
>
> ---
> ### **Comparisons with Additional Baselines**
>
> We compare against 4 additional recent baselines: **VQ-BeT**, **BESO**, **ADPro**, and **Diffusion-ES**. All methods are evaluated on test-time adaptation to constraint-induced behavior shifts using a single offline demonstration (no online interaction). Some methods were not designed for this setting, so we make minimal adaptations: VQ-BeT and BESO are conditioned on sparse waypoints from the provided demo; Diffusion-ES uses demo similarity as the reward; ADPro replaces scene-geometry guidance with gradients from the demo-similarity loss. All methods are trained on multimodal datasets and evaluated under the 40-rollout protocol. We exclude DPPO: it requires online interactions and is incompatible with our setting.
>
> **NOTE: Detailed per-domain, per-baseline results are omitted here due to space limitations.** This summary highlights the full set of newly evaluated techniques and their aggregate performance. **Complete results are in Table 1 of the supplementary [PDF](https://drive.google.com/file/d/111cQl6CaOezeX1G0GnV4rGBU40tjpfzr/view).**
> |Task|Scenes|PDP|DP|BC-GMM|BC|IBC|ADPro|Diff-ES|VQ-BeT|BESO|
> |-|-|-|-|-|-|-|-|-|-|-|
> |CloseDrawer|1-3||||||||||
> |PlaceBlock|1-3||||||||||
> |MeatOffGrill|1-3||||||||||
> |PickUpCup|1-3||||||||||
> |**Overall Avg.**||**92.0**|21.5|13.8|23.3|0.0|17.7|19.9|64.6|30.6|
>
> PDP achieves the best overall average performance (92.0%); VQ-BeT is the strongest baseline (64.6%). The gap is largest on Scene 3, where target behaviors are unseen during training: PDP achieves 100.0 (CloseDrawer), 86.7 (PlaceBlock), 91.7 (MeatOffGrill), and 82.5 (PickUpCup); other baselines degrade substantially. These results confirm that PDP outperforms recent latent-conditioned and test-time adaptation baselines in our constraint-shift setting.
>
> ---
> ### **Test-Time Demonstration Requirement**
>
> PDP is not inherently tied to fitting a full demonstration: the behavior latent $z$ can be optimized against other objectives. Please see the section titled **Adaptation from Alternative Supervision** in our response to **Reviewer ZnkH** for a detailed discussion and additional experiments.
>
> ---
> ### **Evaluation on Additional Benchmarks**
>
> We appreciate this suggestion. Please see our response to **Reviewer MGRm** for a thorough evaluation on new benchmarks, including RoboMimic, Franka Kitchen, and D3IL. Results show that PDP’s advantage holds across 8 domains and 24 scenes, including benchmarks with both designed and naturally occurring multimodality.
>
> ---
> ### **Distortion Bound**
>
> One could assume bi-Lipschitz regularity: a coordinate chart exists that embeds each mode's trajectories into $\mathbb{R}^{d_m}$, preserving distances up to constants $c_m$ and $C_m$, i.e., with distortion $D\le C_m/c_m$ when $d_z\ge d_m$. In robotics, within-mode variation is often low-dimensional: $d_m\in [2,5]$; Calinon et al (2007). This aligns with our experiments: geometry loss is 0.0081 when $d_z=2$, implying a 9% distance error and $D\approx 1.2$ (near the optimum 1.0). Thus, both theory and experiments indicate that when $d_z\ge d_m$, geometry is preserved with low distortion. Sharp bounds for our specific $E_\phi$ are left for future work.
>
> ---
> ### **Mode Collapse & Latent Dimensionality**
>
> The reviewer raised two questions:
>
> **(1)** *How much information may be lost in low-dimensional trajectory embeddings*
>
> First, note that $z$ is not required to encode full trajectories across all modes simultaneously. It acts as a behavior selector: a compact handle that tells the denoiser which mode to execute. The diffusion denoiser retains the within-mode expressive power needed to generate the detailed action sequence. Thus, $d_z$ needs only reflect the degree of behavioral variation the latent must distinguish, rather than the full trajectory or state-action dimensionality.
>
> For a detailed discussion, including the requested sensitivity analysis of $d_z$, please refer to the section titled **Scalability of the Behavior Latent Dimension** in our response to **Reviewer ZnkH**.
>
> **(2)** *Does the latent space preserve mode separation?*
>
> We do not claim a formal guarantee against mode collapse from the VAE objective; our evidence is empirical. Fig.3 of the supplementary PDF shows that training modes form tight, clearly separated clusters in the learned manifold, arranged according to the underlying trajectory geometry, as encouraged by $\mathcal{L}\text{geo}$. Removing geometry alignment substantially degrades adaptation (Table 4), confirming that $\mathcal{L}\text{geo}$ prevents collapse.
>
> ---
> We hope our responses and comparisons with new algorithms and benchmarks address your main concerns. If so, we would appreciate your reconsidering your score, and we welcome any further questions.

---

> > ### Author Rebuttal · Reviewer_wwsZ · 2026-04-03
> >
> > Thank you for the author's reply. Most of my questions have been resolved. The discussion of the relevant work should also be added to the revised manuscript to demonstrate the author's understanding of the relevant work.

---

> > > ### Author Response · Authors · 2026-04-05
> > >
> > > Thank you for this follow-up. Due to the rebuttal space limit, we were not able to elaborate on the final related-work point in the rebuttal itself. We therefore clarify that point below.
> > >
> > > The papers raised by the reviewer will be discussed explicitly in the revised manuscript. In the related-work section, we will add them to the paragraph that currently discusses RL-guided training for diffusion models and control: **DIPO [1]** as an early diffusion-policy formulation for online RL, **QSM [2]** as a representative Q-guided diffusion-policy method, and the two **DACER [3][4]** papers as representative actor-critic and efficiency-oriented advances for online diffusion-policy training. We will then make the distinction explicit that PDP is complementary to this line of work: these methods optimize policy parameters during training, whereas PDP studies controllable test-time steering of a frozen diffusion policy through a geometry-aligned latent behavior space.
> > >
> > > Please let us know if the reviewer still has concerns; we are open to further discussion.
> > >
> > > ---
> > >
> > > [1] Yang L, Huang Z, Lei F, et al. Policy representation via diffusion probability model for reinforcement learning[J]. arXiv preprint arXiv:2305.13122, 2023.
> > >
> > > [2] Psenka M, Escontrela A, Abbeel P, et al. Learning a diffusion model policy from rewards via q-score matching[J]. arXiv preprint arXiv:2312.11752, 2023.
> > >
> > > [3] Wang Y, Wang L, Jiang Y, et al. Diffusion actor-critic with entropy regulator[J]. Advances in Neural Information Processing Systems, 2024, 37: 54183-54204.
> > >
> > > [4] Wang Y, Wang L, Tan M, et al. Enhanced dacer algorithm with high diffusion efficiency[J]. arXiv preprint arXiv:2505.23426, 2025.

---

### Official Review · Reviewer_XeJU · 2026-03-08

**Soundness:** 2
**Presentation:** 3
**Significance:** 3
**Originality:** 3
**Overall Recommendation:** 3
**Confidence:** 5

**Summary:**

The paper introduces Parameterized Diffusion Policy (PDP), a framework that conditions a diffusion policy on a continuous, geometry-aligned latent variable to enable reliable behavior steering. Unlike standard diffusion policies that rely on high-dimensional stochastic noise for diversity, PDP learns a structured behavior manifold where the Euclidean distance between latents correlates with the physical similarity of trajectories. This is achieved using a trajectory encoder trained with a soft-DTW geometry alignment loss. By utilizing global modulation to integrate the latent variable, the model simplifies the denoising process by resolving mode ambiguity. At test time, PDP allows for rapid adaptation to constraint-induced behavior shifts by optimizing only the low-dimensional latent variable via gradient descent, without updating the base policy weights.

**Compliance With Llm Reviewing Policy:**

Affirmed.

**Final Justification:**

Thank you for the clarification regarding Table 3. I acknowledge the real-robot results exist; my concern is their statistical power (n=5 per condition) and scope (single task). Regarding conditioning, I appreciate the precedents cited, but those works validate across far broader settings. My core concern remains: the method's practical value is limited by requiring a successful demonstration under novel constraints — the very thing deployment aims to avoid. I keep my score from 4 (Weak Accept) to 3 (Weak Reject).

**Key Questions For Authors:**

1. The test-time latent adaptation process heavily relies on a new target demonstration $\tilde{\tau}$ to compute the fitting objective $\mathcal{L}_{fit}$ and the warm-start $z_0$. How does the framework perform if only a partial demonstration (e.g., just the altered segment of the trajectory) or a non-kinematic cost function (e.g., obstacle collision penalty) is provided?

2. Table 5 demonstrates that global modulation significantly outperforms naive concatenation. Given that many conditional diffusion models successfully use concatenation or cross-attention, why does concatenation fail so severely (e.g., 25.0% success on PickUpCup) in your specific architecture?

3. The soft-DTW loss is computed on "representative features $X(\tau)$ (e.g., positional states)". How sensitive is the geometry alignment to the selection of these features? Does the inclusion of higher-dimensional data, such as orientation or continuous gripper states, destabilize the metric?

4. You mentioned running $S$ steps of gradient descent before execution. How does the inference latency of running 10-100 optimization steps  affect the feasibility of deploying this method in highly dynamic or reactive real-world scenarios?

**Limitations:**

The authors adequately discuss the limitations of existing methods (e.g., ill-conditioned noise-space steering), but they do not explicitly detail the limitations of their own proposed approach.
Constructive suggestion: Include a formal limitations paragraph discussing the strict dependency on a test-time demonstration for adaptation and the computational overhead introduced by the gradient-based latent fitting process during inference.

**Strengths And Weaknesses:**

Soundness: The submission is technically robust. The theoretical decomposition of variance in Section A.4 provides a compelling mathematical justification for why explicit mode disambiguation provides training relief for the denoiser. The choice of soft-DTW  is highly appropriate to enforce time-invariant geometric alignment. However, a notable weakness is the reliance on a "semantic warm-start" using a newly provided target demonstration at test time. This requirement somewhat dilutes the autonomous adaptation claim, as the system relies on an external oracle to provide a valid trajectory under new constraints.

Presentation: The paper is exceptionally well-structured and clearly written. The distinction between standard observation-side distribution shifts and constraint-induced behavior shifts is sharply articulated in the introduction. Furthermore, the visualizations of the latent space (Figure 5 and Figure 12) effectively demonstrate the continuous and navigable nature of the learned manifold.

Significance: The problem addressed—steerability and adaptability of multimodal behavior cloning models under changing environmental constraints—is a critical bottleneck in deploying robot learning systems. By transforming diffusion from a mechanism of stochastic diversity into a precise tool for behavior steering, this work offers substantial practical utility for the robotics community.

Originality: While individual components (diffusion policies, Soft-DTW, FiLM conditioning) are established, their synthesis into the PDP framework to explicitly solve constraint-induced shifts is highly original. The paper successfully extends classical ideas of parameterized skills into the modern generative diffusion paradigm.

---

> ### Author Rebuttal · Authors · 2026-03-31
>
> We thank the reviewer for their positive assessment of our paper and for their encouraging comments on the significance of the problem, the originality of our approach, and the soundness of our method. Below, we address their questions.
>
> ---
> ### **Adaptation from Alternative Supervision**
>
> We thank the reviewer for raising this practical question. PDP is not inherently tied to fitting a full demonstration: the behavior latent $z$ is a continuous parameter that can be optimized against other objectives as well. We kindly refer the reviewer to our response to **Reviewer ZnkH**, who raised a related question, for a detailed discussion and additional experiments.
>
> ---
> ### **Why Global Modulation Outperforms Concatenation**
>
> Thank you for this question. The key difference lies in the **role and dimensionality** of the conditioning signal $z$ relative to what other conditional diffusion models typically use.
>
> In most conditional diffusion models, the conditioning signal is high-dimensional and information-rich: it carries detailed spatial, semantic, or temporal content. For example, SR3 conditions super-resolution by concatenating an upsampled low-resolution image with the noisy input [15], while Latent Diffusion Models inject text or layout conditions through cross-attention [16]. In those settings, standard conditioning mechanisms work because the conditioning signal itself is already strong during training.
>
> In PDP, by contrast, the behavior latent $z$ is deliberately **low-dimensional** and encodes only the degrees of freedom needed to distinguish behavioral modes, not the full content of the trajectory. When this compact signal is concatenated to a high-dimensional noisy input $A_t^k$ (which has hundreds of dimensions), the network can easily learn to ignore it during training, especially early on when the denoiser loss is dominated by within-mode noise prediction. This is precisely what we observe: the Concatenation variant converges to a training loss nearly identical to the Unconditioned baseline (Appendix C.1, Figure 9), indicating the network learned to disregard $z$ entirely.
>
> **Global modulation** avoids this by a fundamentally different integration mechanism. Instead of adding $z$ as an extra input feature, it applies layer-wise affine transformations $h^{(\ell)} \leftarrow \gamma^{(\ell)}(z) \odot h^{(\ell)} + \beta^{(\ell)}(z)$ to intermediate feature maps at every layer. The network *cannot* ignore $z$ because it directly modulates the computation at every layer, converting the denoiser from a global mixture-modeling problem into a mode-specific denoising problem and resulting in an order-of-magnitude reduction in training loss (Appendix C.1, Figure 9).
>
> In summary: concatenation fails not because it is inherently flawed, but because a low-dimensional mode selector is easily drowned out when appended to a much larger input. Global modulation forces the network to condition on $z$ at every computational stage, making the mode signal inescapable. This architectural choice is specifically important when the conditioning signal is compact and mode-selecting rather than content-rich.
>
> ---
> ### **On the Choice of Behavioral Features $X$**
>
> Thank you for raising this point. We kindly refer the reviewer to our response to **Reviewer ZnkH**, who raised a related question, for a full discussion.
>
> ---
> ### **On Inference Latency of Test-Time Optimization**
>
> Thank you for this question. This is indeed an important practical consideration. The $S$ optimization steps are a *one-time adaptation cost* incurred when the scene or constraint changes, not a per-action latency: once the optimized $z^{\*}$ is obtained, it remains fixed throughout the rollout, and each control step runs standard diffusion denoising conditioned on
> $z^{\*}$ at the same speed as a standard diffusion policy. In timing measurements on a single NVIDIA 2080 Ti, per-action execution remains approximately 7-15 ms, while the one-time fitting cost is approximately 0.1 s for 10 gradient steps and 0.5 s for 50 steps. In the manipulation settings studied here, scene or constraint changes occur at the task level rather than at control frequency, so this one-time adaptation overhead remains practical even for time-sensitive deployments.
>
> ---
> We hope our responses have addressed your main concerns. We will incorporate all discussed clarifications into the final version. The new experimental results shared in response to other reviewers are also available for your consideration. If these address the main points you raised, we would appreciate your reconsidering your score and welcome any further questions.

---

> > ### Author Rebuttal · Reviewer_XeJU · 2026-04-03
> >
> > Thank you for the detailed responses. However, after reflection, two concerns lead me to lower my score.
> >
> >
> > First, the explanation of why concatenation fails, while insightful, actually exposes an architectural fragility. By the authors' own analysis, PDP fundamentally cannot work with the most common conditioning mechanisms in the diffusion literature (concatenation, cross-attention) — it only succeeds with FiLM-style modulation. This tight coupling to a specific architectural choice makes the contribution feel more like a narrow design recipe than the generalizable framework the paper claims to be.
> > Second, for a paper that emphasizes practical deployment and utility for the robotics community, the complete absence of real-world experiments remains a significant gap. Simulation timing numbers do not address real-world complexities such as sensor noise, calibration errors, and sim-to-real transfer. Even a minimal proof-of-concept on a real robot would have substantially strengthened the claims.
> >
> >
> > I am revising my score from 4 (Weak Accept) to 3 (Weak Reject).

---

> > > ### Author Response · Authors · 2026-04-05
> > >
> > > Thank you for pointing out these concerns. We would like to clarify them further here and are open to further discussion.
> > >
> > > ---
> > >
> > > ## On Real-Robot Experiments
> > >
> > > We would like to clarify that the paper does **already include real-world experiments in the original submission**, together with quantitative results. These results are reported in Table 3 on page 7, discussed in Section 5.2, and the real-robot setup and dataset construction details are provided in Appendix B.1.
> > >
> > > In addition, footnote 1 on page 7 links to videos of all real-robot experiments. The submission, therefore, does include both a real-robot proof-of-concept and quantitative hardware results.
> > >
> > > ---
> > >
> > > ## On the Conditioning Mechanism
> > >
> > > We appreciate the reviewer's comments. That said, please note that our paper **does not claim conditioning-strategy invariance**. In particular, we do not claim that PDP should work equally well with every possible conditioning mechanism, nor that our contribution is conditioning-strategy agnostic. **On the contrary**, one of the main design contributions is precisely the study and demonstration that FiLM-style global modulation is **necessary** to support a compact behavior latent space. These conclusions are supported by the explicit ablation in Table 5 and Appendix C.1. Global modulation is indeed one of the key architectural design choices that allows PDP to outperform all competitor baselines across all benchmark problems. Our ablation was included precisely to test that design choice in the low-dimensional behavior-latent setting investigated in the paper.
> > >
> > > More broadly, we do not agree that it is accurate or fair to interpret this design choice as a unique fragility of PDP. **Tailoring the conditioning interface to the structure of the conditioning signal is common practice.** Prior work often treats the conditioning mechanism as part of the method itself rather than an incidental implementation detail. FiLM introduced feature-wise affine modulation and validated it through architectural ablations [1]; AdaIN showed that adaptive normalization fuses content and style more effectively than simple concatenation [2]; SPADE showed that intermediate-layer concatenation underperforms spatially adaptive modulation [3]; Latent Diffusion Models use different conditioning routes for different modalities, using cross-attention for text-like conditioning and concatenation for spatially aligned inputs [4]; and DiT directly compared in-context conditioning, cross-attention, and adaptive layer normalization, finding adaLN-Zero best [5]. These precedents support a narrower conclusion than conditioning-strategy invariance: **the best conditioning interface depends on the structure of the conditioning variable.**
> > >
> > > Our point here is therefore specific and constructive: for the type of conditioning variable studied in PDP, namely a compact latent that selects among behavior modes rather than carrying rich observation content, global modulation is a key architectural ingredient. This is exactly why we included the concatenation comparison in Table 5.
> > >
> > > On a more general note, although we do understand the reviewer's comment on this point, we believe that arguing that our method would not have outperformed all baselines, $\underline{\text{had we used a different architecture than the one we actually proposed}}$ does not constitute a substantive criticism (it is, in fact, one that could be raised about virtually any paper). We strongly believe that a paper should be evaluated based on the merits of the methods it actually introduces and evaluates, rather than on speculation about how alternative design choices might have performed.
> > >
> > > To further clarify this point, we have included an additional discussion in the appendix that carefully examines these design choices and the potential impact of replacing our proposed architecture with alternative conditioning mechanisms.
> > >
> > > ---
> > >
> > > **References**
> > >
> > > [1] Perez et al., "FiLM: Visual Reasoning with a General Conditioning Layer," AAAI 2018.
> > >
> > > [2] Huang & Belongie, "Arbitrary Style Transfer in Real-Time With Adaptive Instance Normalization," ICCV 2017.
> > >
> > > [3] Park et al., "Semantic Image Synthesis With Spatially-Adaptive Normalization," CVPR 2019.
> > >
> > > [4] Rombach et al., "High-Resolution Image Synthesis With Latent Diffusion Models," CVPR 2022.
> > >
> > > [5] Peebles & Xie, "Scalable Diffusion Models with Transformers," ICCV 2023.

---

### Official Review · Reviewer_MGRm · 2026-03-09

**Soundness:** 2
**Presentation:** 3
**Significance:** 2
**Originality:** 3
**Overall Recommendation:** 3
**Confidence:** 4

**Summary:**

This paper proposes Parameterized Diffusion Policy (PDP), which extends diffusion policies with a continuous behavior latent z learned via soft-DTW geometry alignment. The latent is injected into the denoiser through FiLM-style global modulation. At test time, PDP adapts to constraint shifts by optimizing z alone (frozen weights), warm-started from an encoder. Experiments on four RLBench simulation tasks and one real-robot task show improvements over standard DP, BC, BC-GMM, and IBC. Overall, the authors analyze the context of constraint-induced behavior shifts as a distinct challenge for diffusion policies. The paper considers an important domain in robotic behavior adaptation.

**Compliance With Llm Reviewing Policy:**

Affirmed.

**Key Questions For Authors:**

1.	How does PDP compare against recent latent-conditioned methods (BESO, VQ-BeT, Latent Actions)? This is the most important question—a positive answer could change my score toward acceptance.
2.	Can the authors provide photographs or video frames of actual robot execution across all evaluation conditions? The current lack of visual evidence is a serious credibility concern.
3.	How sensitive is PDP to the choice of behavior trace features X(τ)? Would it work for tasks where modes differ in force or timing rather than position?
4.	Can PDP adapt from weaker supervision (partial trajectory, goal image, cost function) rather than requiring a full demonstration?

**Limitations:**

The paper lacks a dedicated limitations section. Key unaddressed limitations include: scalability of d_z = 2 to richer behavior diversity, reliance on full demonstrations for adaptation, gap between artificial and natural multimodal datasets, and safety implications of synthesizing novel untested behaviors via latent interpolation.

**Strengths And Weaknesses:**

Strengths
1.	Well-motivated problem. The distinction between observation-side shift and constraint-induced behavior shift (Figure 1) is insightful and clearly articulated.
2.	Principled geometry alignment. Using differentiable soft-DTW to align latent distances with trajectory similarity is well-motivated and produces clearly structured latent spaces (Figures 5, 12–14).
3.	Informative ablations. Tables 4–6 cleanly isolate contributions of geometry alignment, global modulation, and encoder warm-start respectively.
4.	Strong zero-mode results. PDP achieves 82.5–100% in Scene 3 (novel behavior discovery) where all baselines nearly completely fail, demonstrating genuine behavior synthesis beyond memorized modes.
Weaknesses
1.	Severely outdated baselines. The primary comparison is against standard DP (Chi et al., 2023), which is nearly three years old. Other baselines (BC, BC-GMM, IBC) are even older. The paper lacks comparison with recent latent-conditioned methods (BESO, VQ-BeT, Latent Actions) and adaptive diffusion approaches (DPPO, AdPro). This makes it impossible to assess whether PDP advances beyond the current state of the art.
2.	No visual evidence of real-robot execution. The real-robot experiment covers only one task (OpenDrawer, 5 trials). More critically, neither the main text nor the appendix contains any photograph or video frame of the robot actually performing tasks. Figure 8 only shows the setup with obstacles; Figure 11 shows trajectory curves, not execution images. Even the original DP paper from 2023 provided extensive real-robot visual documentation. This omission seriously undermines the credibility of the real-world claims.
3.	Artificial multimodal datasets. Modes are explicitly designed (e.g., 8 approach angles, 4 grasp points) with uniform coverage. Natural demonstrations exhibit continuous variability and ambiguous mode boundaries. No evaluation on standard benchmarks (RoboMimic, CALVIN) with naturally occurring multimodality.
4.	Limited scalability evidence. All tasks use d_z = 2. It is unclear whether PDP scales to higher-dimensional behavior spaces or tasks where positional DTW features are insufficient (e.g., force-sensitive or timing-critical tasks).
5.	Adaptation requires a full demonstration. Test-time fitting needs a complete successful trajectory under new constraints. Obtaining such a demonstration may be as hard as solving the task itself, limiting practical applicability.
6.	No failure analysis. PDP achieves 73.3% on MeatOffGrill (Table 1) and 82.5% on PickUpCup Scene 3 (Table 2). The paper does not analyze what causes the remaining failures.

---

> ### Author Rebuttal · Authors · 2026-03-31
>
> We thank the reviewer for the positive comments on our method’s theoretical soundness, ablations, and zero-shot results. The reviewer's questions focused on 3 aspects: **(1)** additional baselines; **(2)** visual documentation of real-robot experiments; and **(3)** evaluation on further benchmarks. Below, we address each of these points, as well as additional questions raised.
>
> ---
> ### **Evaluation on Additional Benchmarks**
>
> We appreciate this suggestion. We have added experiments on 4 new domains: **Door** (RoboMimic), **Microwave** (Franka Kitchen), and **Avoiding 24** and **Avoiding 32** (D3IL). The original Door and Microwave datasets are largely unimodal; this is consistent with recent findings that many standard robot-learning benchmarks have limited multimodality [14]. Thus, we augmented each with 3 additional approach modes (4 total). We train on all 4 modes and evaluate on 3 constraint-shifted scenes. Avoiding 24 and 32 are standardized D3IL tasks with naturally multimodal demonstrations via distinct obstacle-avoidance corridors [1]; here, we train on 20 modes and use the same 3-scene protocol.
>
> **NOTE: Detailed per-domain, per-baseline results are omitted here due to space limitations.** *This summary highlights the full set of newly evaluated techniques, the new benchmarks, and the aggregate performance.* **Complete results are in Table 2 of the supplementary [PDF](https://drive.google.com/file/d/111cQl6CaOezeX1G0GnV4rGBU40tjpfzr/view).**
> |Task|Scenes|PDP|DP|BC-GMM|BC|IBC|ADPro|Diff-ES|VQ-BeT|BESO|
> |-|-|-|-|-|-|-|-|-|-|-|
> |Door|1-3||||||||||
> |Microwave|1-3||||||||||
> |Avoiding 24|1-3||||||||||
> |Avoiding 32|1-3||||||||||
> |**Overall Average**||**98.0**|18.2|16.7|33.7|0.0|21.7|65.3|21.5|12.2|
>
> PDP achieves 98.0% overall. Note that the D3IL Avoiding tasks directly address the reviewer's concern about natural multimodality: these have continuous variability and no artificially imposed mode structure, and PDP achieves near-perfect performance across all its corresponding scenes. Diff-ES performs well on in-distribution modes but drops to 0% on Avoiding 32's unseen mode; this highlights the limits of noise-space optimization without a structured behavior manifold. VQ-BeT performs well on Microwave but fails on Door & D3IL Avoiding. Together with Table 2, these results show that PDP’s advantage holds across 8 domains and 24 scenes, including benchmarks with both designed and naturally occurring multimodality.
>
> ---
> ### **Comparisons with Additional Baselines**
>
> We thank the reviewer for this suggestion and kindly refer them to our detailed response to **Reviewer wwsZ**, who raised a related question. There, we provide a full discussion and a thorough comparison with four recent baselines: **VQ-BeT**, **BESO**, **ADPro**, and **Diffusion-ES**. These new comparisons confirm that PDP outperforms recent latent-conditioned and test-time adaptation methods in our constraint-shift setting.
>
> ---
> ### **Visual Documentation of Robot Experiments**
>
> Thank you for this comment. We will ensure that the link to our video website (provided in footnote 1, page 7) is more prominently displayed in the updated version of the paper, possibly using a different color. The [website](https://sites.google.com/view/parameterized-dp) includes footage of real-robot executions across all evaluation conditions (original scene & constraint-shifted scenes). We will also add representative video frames in the appendix.
>
> ---
> We thank the reviewer for raising the following 3 questions:
>
> 1. **Scalability of the Behavior Latent Dimension**
> 2. **On the Choice of Behavioral Features $X$**
> 3. **Adaptation from Alternative Supervision**
>
> Please see our responses to **Reviewer ZnkH**, who raised related questions, for a full discussion.
>
> ---
> ### **Failure Analysis**
>
> The remaining failures in MeatOffGrill and PickUpCup Scene 3 mainly stem from precision at the grasp stage. This issue can affect all tasks, but it is more pronounced here since graspable targets are smaller and less forgiving than in other tasks: the cup grasp must align with a small handle region on the rim, and the meat-grasping stage is sensitive to small approach errors, especially after rerouting around new obstacles. This is further amplified in MeatOffGrill due to the task's sequential structure, where an imperfect grasp can propagate error into the downstream transport phase. Despite these failures, PDP substantially outperforms *all* baselines on both tasks across constraint-shifted scenes (including 84.2-91.7% on MeatOffGrill, where most other methods are near 0%). We will add this analysis to the revised appendix.
>
> ---
> We hope our responses, including the new comparisons with BESO, VQ-BeT, ADPro, and Diff-ES and the additional benchmarks, address your main concerns. We will incorporate all revisions into the final version. If these clarifications and results resolve the points you raised, we would appreciate your reconsidering your score and welcome any further questions

---

> > ### Author Rebuttal · Reviewer_MGRm · 2026-04-02
> >
> > Two core concerns remain unresolved.
> > First, the rebuttal mentions real-robot experiments but provides no quantitative results (success rates, number of trials, experimental setup). Video footage alone is insufficient.
> > Second, the updated baseline comparison table in the rebuttal contains empty cells for all per-task results. The authors refer to a supplementary PDF for complete data, but I was unable to locate it. A rebuttal should present key evidence directly and accessibly. I can only treat these comparisons as unsubstantiated.
> > I acknowledge the rebuttal and maintain my current score.

---

> > > ### Author Response · Authors · 2026-04-05
> > >
> > > Thank you for the follow-up. In the rebuttal, we did include the supplementary PDF containing the complete per-task results; because rebuttal space was limited, **the link was embedded on the word "PDF."** For clarity, we paste the direct link here as plain text:
> > >
> > > https://drive.google.com/file/d/111cQl6CaOezeX1G0GnV4rGBU40tjpfzr/view
> > >
> > > We would like to further clarify two points below.
> > >
> > > ---
> > >
> > > ## Quantitative Results Real Robot Experiments
> > >
> > > The quantitative real-robot results — including success rates, number of trials, and experimental setup — are already reported in the **original submission** (Line 371, Table 3, and Appendix B.1 at Line 874). Table 3 of the original paper reports 5 trials in each of the 4 evaluated scenes, and we paste those results here for your convenience.
> > >
> > > |Method|Original|Scene 1|Scene 2|Scene 3|
> > > |-|-|-|-|-|
> > > |PDP|**5/5**|**5/5**|**5/5**|**5/5**|
> > > |DP|5/5|3/5|1/5|2/5|
> > >
> > > ---
> > >
> > > ## Complete Results for the Additional Baselines and Benchmarks
> > >
> > > In our rebuttal, we added the additional comparisons raised in the review: **four recent baselines** (VQ-BeT, BESO, ADPro, Diff-ES) and **four additional benchmark domains** (Door, Microwave, Avoiding 24, Avoiding 32). For your convenience, we include those full tables directly below.
> > >
> > > Table A. Full results on the original benchmark with the added baselines (CloseDrawer, PlaceBlock, MeatOffGrill, PickUpCup):
> > >
> > > |Task|Scene|PDP|DP|BC-GMM|BC|IBC|ADPro|Diff-ES|VQ-BeT|BESO|
> > > |-|-|-|-|-|-|-|-|-|-|-|
> > > |CloseDrawer|1|**100.0±0.0**|**100.0±0.0**|0.0±0.0|0.0±0.0|0.0±0.0|**100.0±0.0**|99.2±0.7|**100.0±0.0**|92.5±2.5|
> > > ||2|**100.0±0.0**|88.3±14.2|0.0±0.0|82.5±4.3|0.0±0.0|96.7±1.8|34.2±2.5|**100.0±0.0**|97.5±2.5|
> > > ||3|**100.0±0.0**|40.0±13.0|0.0±0.0|2.5±2.5|0.0±0.0|0.0±0.0|1.7±0.7|**100.0±0.0**|**100.0±0.0**|
> > > |PlaceBlock|1|**95.0±2.5**|0.0±0.0|0.0±0.0|12.5±3.8|0.0±0.0|0.0±0.0|14.2±4.8|90.0±2.5|0.0±0.0|
> > > ||2|**95.8±1.4**|0.0±0.0|0.0±0.0|2.5±2.5|0.0±0.0|0.0±0.0|13.3±5.4|92.2±0.9|22.5±15.9|
> > > ||3|86.7±10.1|2.5±2.5|0.0±0.0|5.0±2.5|0.0±0.0|0.0±0.0|13.3±5.4|76.0±3.1|16.2±11.5|
> > > |MeatOffGrill|1|**84.2±1.4**|0.0±0.0|0.0±0.0|80.0±5.0|0.0±0.0|0.0±0.0|0.0±0.0|0.0±0.0|0.0±0.0|
> > > ||2|**86.7±5.2**|0.0±0.0|0.0±0.0|65.0±3.8|0.0±0.0|0.8±0.7|0.0±0.0|7.5±2.5|0.0±0.0|
> > > ||3|**91.7±2.9**|2.5±2.5|0.0±0.0|0.0±0.0|0.0±0.0|0.8±0.7|2.5±2.5|0.0±0.0|0.0±0.0|
> > > |PickUpCup|1|**95.8±1.4**|7.5±13.0|92.5±2.5|2.5±2.5|0.0±0.0|0.0±0.0|20.0±4.1|85.0±2.5|0.0±0.0|
> > > ||2|**85.0±2.5**|0.8±1.4|72.5±4.3|4.2±2.9|0.0±0.0|10.8±4.9|20.0±3.5|55.0±5.0|22.5±15.9|
> > > ||3|**82.5±4.3**|15.8±21.0|0.0±0.0|22.5±4.3|0.0±0.0|3.3±1.4|20.0±2.5|70.0±2.5|16.2±11.5|
> > > |**Overall Average**||**92.0**|21.5|13.8|23.3|0.0|17.7|19.9|64.6|30.6|
> > >
> > > Table B. Full results on the four additional benchmark domains added in response to the review (Door, Microwave, Avoiding 24, Avoiding 32):
> > >
> > > |Task|Scene|PDP|DP|BC-GMM|BC|IBC|ADPro|Diff-ES|VQ-BeT|BESO|
> > > |-|-|-|-|-|-|-|-|-|-|-|
> > > |Door|1|**98.3±1.7**|22.5±3.8|**100.0±0.0**|0.0±0.0|0.0±0.0|25.0±2.9|92.0±3.5|0.0±0.0|0.0±0.0|
> > > ||2|96.7±1.7|39.2±5.8|**100.0±0.0**|78.0±4.1|0.0±0.0|51.7±2.2|90.0±2.5|0.0±0.0|0.0±0.0|
> > > ||3|**87.5±12.3**|19.2±0.8|0.0±0.0|0.0±0.0|0.0±0.0|16.7±0.8|62.0±5.7|0.0±0.0|0.0±0.0|
> > > |Microwave|1|**98.3±1.2**|12.5±1.4|0.0±0.0|6.7±2.4|0.0±0.0|17.5±2.0|16.7±4.2|61.7±3.1|35.0±10.2|
> > > ||2|**100.0±0.0**|45.0±12.7|0.0±0.0|18.3±2.4|0.0±0.0|61.7±3.1|46.7±5.1|**100.0±0.0**|60.8±8.5|
> > > ||3|**100.0±0.0**|27.5±3.5|0.0±0.0|1.7±1.2|0.0±0.0|45.0±2.0|36.7±6.6|96.7±2.4|50.0±7.4|
> > > |Avoiding 24|1|**100.0±0.0**|13.8±2.4|0.0±0.0|33.3±4.7|0.0±0.0|11.2±5.5|**100.0±0.0**|0.0±0.0|0.0±0.0|
> > > ||2|**100.0±0.0**|8.8±1.4|0.0±0.0|33.3±2.4|0.0±0.0|5.0±1.4|**100.0±0.0**|0.0±0.0|0.0±0.0|
> > > ||3|**100.0±0.0**|0.0±0.0|0.0±0.0|33.3±5.9|0.0±0.0|0.8±1.2|39.2±2.4|0.0±0.0|0.0±0.0|
> > > |Avoiding 32|1|**100.0±0.0**|16.2±3.1|0.0±0.0|66.7±3.1|0.0±0.0|11.2±4.7|**100.0±0.0**|0.0±0.0|0.0±0.0|
> > > ||2|**100.0±0.0**|13.8±2.4|0.0±0.0|66.7±4.7|0.0±0.0|15.0±3.5|**100.0±0.0**|0.0±0.0|0.0±0.0|
> > > ||3|**95.8±1.2**|0.0±0.0|0.0±0.0|66.7±2.4|0.0±0.0|0.0±0.0|0.0±0.0|0.0±0.0|0.0±0.0|
> > > |**Overall Average**||**98.0**|18.2|16.7|33.7|0.0|21.7|65.3|21.5|12.2|
> > >
> > > With these comparisons to the additional recent baselines and benchmark domains raised in the initial review, the results further support our main claim: across both the original benchmark and the additional domains, PDP remains the strongest overall method. These results support the paper’s central claim that a geometry-aligned behavior latent provides a more effective interface for constraint-shift adaptation than standard diffusion baselines and recent latent-conditioned alternatives. We thank the reviewer for the suggestion to broaden the empirical comparisons.
> > >
> > > ---
> > >
> > > We welcome any further questions and discussion. If these clarifications and results address the remaining concerns, we would be grateful if the reviewer would consider updating the score.

---

### Official Review · Reviewer_ZnkH · 2026-03-12

**Soundness:** 4
**Presentation:** 3
**Significance:** 3
**Originality:** 4
**Overall Recommendation:** 5
**Confidence:** 4

**Summary:**

The paper introduces Parameterized Diffusion Policies (PDP), a framework designed to improve the controllability and test-time adaptability of diffusion-based behavior cloning in robotic manipulation. While standard diffusion policies are excellent at modeling multimodal distributions, they rely on stochastic noise for diversity, making them difficult to steer when environmental constraints change (e.g., a new obstacle blocks a previously valid path). To solve this, PDP maps demonstration trajectories into a continuous, low-dimensional latent space using a trajectory encoder. Crucially, the authors use a Soft-DTW (Dynamic Time Warping) geometry loss to ensure that distances in the latent space correspond to physical trajectory similarities. The diffusion model is then conditioned on this latent space using a global modulation architecture. At test time, if the environment changes, PDP can quickly adapt without updating the policy weights by running gradient descent directly on the behavior latent, initialized by a new target demonstration. The method is evaluated on both simulated and real-robot tasks, showing significant improvements over standard diffusion policies and behavior cloning baselines.

**Compliance With Llm Reviewing Policy:**

Affirmed.

**Final Justification:**

Keep my score as all the concerns are addressed.

**Key Questions For Authors:**

1. For the constraint-shifted scenes, the adaptation mechanism requires a target demonstration (a "semantic warm-start") to optimize the latent. Can authors discuss how the system might perform if adaptation was guided by a cost function or safety constraint instead of requiring a human-provided trajectory?

**Limitations:**

Potential limitations are listed in weaknesses.

**Strengths And Weaknesses:**

## Strengths
- It addresses a well-motivated problem.
- Clever Latent Structuring: Using Soft-DTW to align the latent geometry with actual physical trajectory similarity is an elegant solution. It directly enables smooth interpolation between modes and prevents the topological distortions common in standard VAEs.
- Strong Empirical Results: PDP demonstrates impressive performance margins over baselines, particularly in the difficult "zero-mode feasible" scenarios where the robot must discover a novel behavior.
## Weaknesses
- The paper fixes the behavior latent dimension at $d_z = 2$ throughout all experiments (Table 9). While this choice aids visualization and simplifies test-time optimization, it raises a scalability concern in tasks with large mode counts (e.g., 64 modes in PLACEBLOCK and MEATOFFGRILL). Faithfully preserving pairwise Soft-DTW distances across a large number of geometrically distinct trajectories in a 2-dimensional space is topologically challenging, and the paper does not include an ablation studying how performance varies with $d_z$ across tasks of differing modal complexity. An analysis of this trade-off would strengthen confidence in the method's broader applicability.
- The geometry alignment loss $\mathcal{L}_{geo}$ is computed over end-effector positional traces $X(\tau)$. As a result, trajectories that share identical spatial paths but differ in gripper state, contact configuration, or applied force will be assigned a Soft-DTW distance of zero and mapped to the same point in the latent space. This limits the expressiveness of the behavior manifold to geometric variation in end-effector position, potentially excluding physically meaningful behavioral distinctions. The paper would benefit from a brief discussion of this design choice and its implications for tasks involving rich contact interactions (involving rich gripper action changes, e.g. Pick-and-place for many times in one task) or non-geometric behavioral variation.

---

> ### Author Rebuttal · Authors · 2026-03-31
>
> We thank the reviewer for their positive assessment of our paper and for their encouraging comments on the relevance of the problem, the strength of our method, and the quality of our experiments. Below, we address their questions:
>
> ---
> ### **Scalability of the Behavior Latent Dimension**
>
> Thank you for raising this point.
>
> **On whether $d_z = 2$ is sufficient for tasks with many modes**. First, note that in PDP, $z$ is not meant to encode full trajectories across all modes simultaneously. It acts as a *behavior selector*: a compact handle that tells the denoiser which mode to execute. In long-horizon tasks like PlaceBlock and MeatOffGrill, the 64 modes arise from 8 reaching strategies crossed with 8 carrying strategies. As discussed in Section 4.3, we address this through segmented latents: each demonstration is split into functional phases (reaching & carrying), and an independent latent is fit per phase. Each latent therefore needs only to separate 8 modes, not all 64. Separating 8 compact clusters does not require more than 2 dimensions, as shown in Figs. 5, 12, and 13, where 8 well-separated clusters are clearly identifiable in the learned 2D space. This is why $d_z=2$ remains sufficient even in these tasks, consistent with the strong results in Table 2.
>
> **On the ablation over $d_z$.** To directly address the scalability concern raised by the reviewer, we added an ablation studying how performance varies with $d_z$ on D3IL Avoiding [1], which contains naturally multimodal demonstrations:
> |Dataset|$d_z=2$|$d_z=4$|$d_z=8$|
> |-|-|-|-|
> |Avoiding 24|7.5|**100.0**| **97.5**|
> |Avoiding 32|0.0|67.5|**95.8**|
>
> *PDP's success rate on unseen-mode adaptation. See Table 3 in the supplementary [PDF](https://drive.google.com/file/d/111cQl6CaOezeX1G0GnV4rGBU40tjpfzr/view?usp=sharing).*
>
> These results show that as modal complexity increases, larger latent dimensions may be needed: $d_z=4$ suffices for Avoiding 24, while Avoiding 32 (which requires greater behavioral diversity) requires $d_z=8$. Importantly, the required dimensionality remains modest as task complexity grows, and performance scales cleanly with $d_z$. This confirms that PDP generalizes beyond the $d_z=2$ setting: when tackling complex task families, increasing $d_z$ reliably restores strong performance.
>
> ---
> ### **On the Choice of Behavioral Features $X$**
> The reviewer is correct that in our experiments, since $X(\tau)$ consists of EE positional traces, trajectories with identical spatial paths but different gripper states have zero soft-DTW distance. Two clarifications are important:
> 1. $X$ can be any representative features of a demonstration (see Section 4.1). Our method is agnostic to this choice, and the designer can define $X$ based on the task family. Positional traces are one example, but gripper state, contact forces, or other relevant signals can be included. We'll clarify that EE position was specific to our experiments, not an assumption of the method.
> 2. The choice of EE traces in our experiments was deliberate: in our tasks, the dominant source of inter-mode variation is geometric, and such traces are sufficient to capture these distinctions, consistent with common practice in recent robot learning [2-4]. Importantly, this choice affects only how the latent space is organized: the denoiser is trained on the full action distribution, *including gripper state and all non-positional dimensions*, so behavioral variation not captured by $X$ is handled by the denoiser rather than through the latent structure.
>
> ---
> ### **Adaptation from Alternative Supervision**
>
> Thank you for this question. PDP is not inherently tied to fitting a target demonstration: the behavior latent $z$ is a continuous parameter that can be optimized against any differentiable objective, including cost functions or safety constraints. We focus on demonstration-guided adaptation since it provides a concrete/challenging test of policy adaptation under shifted constraints, without online interaction or weight updates. This setting is closely related to recent work on one- and few-shot imitation & policy adaptation.
>
> To show that PDP also supports cost-based supervision, we added an experiment on CloseDrawer where adaptation is guided purely by a task-level cost, with no target demonstration. Cost is defined as +1 for successful closure, -1 for wall collision, and -0.5 for failure to close. We optimize $z$ using the Cross-Entropy Method (CEM). As shown in Fig.1 of the supplementary PDF, success rate increases from near 0% to 100% within 20 CEM iterations, with trajectories converging from infeasible to collision-free strategies. This experiment was added to address the reviewer’s question and to show that our framework naturally extends to cost-based supervision without modifying the underlying method: this is a promising complementary direction, and we leave a comprehensive analysis to future work.

---

> > ### Author Rebuttal · Reviewer_ZnkH · 2026-04-06
> >
> > Thanks for the response. My questions are totally solved. I will keep my positive rating.

---

### Decision · Program_Chairs · 2026-04-30

**Decision:**

Accept (regular)

**Comment:**

Reviewers were generally positive on the problem considered, as well as the proposed solution.  Clear contributions include the Soft-DTW geometry alignment loss which produces clearly structured latent spaces, the FiLM-based deep modulation architecture that is shown to be necessary through a direct ablation, and the zero-mode results where PDP achieves 82-100% on novel behavior discovery while all baselines nearly fail. In response to reviewer requests, the authors added comparisons against four recent baselines (VQ-BeT, BESO, ADPro, Diffusion-ES) and four additional benchmark domains including naturally multimodal D3IL tasks, and PDP's advantage holds throughout.  As such, we recommend acceptance of the paper.

We emphasize however that despite this generally positive view, two main concerns were raised that should be addressed in the final paper. Reviewer XeJU argued that PDP's dependence on FiLM-style modulation is an architectural fragility, since the method does not generalize to more common conditioning mechanisms such as concatenation or cross-attention. The authors responded that tailoring the conditioning interface to the structure of the conditioning variable is standard practice with precedent across the diffusion literature, and the ablation in Table 5 directly supports the design choice. Given that this concern may be shared by other readers, the paper would benefit from a clearer discussion of this design choice in the main text rather than relegating it to the appendix.

The more substantive issue is that test-time adaptation requires a successful demonstration of the desired behavior under the new constraints, which in some deployment scenarios may be as hard to obtain as solving the task itself. The rebuttal includes a CEM experiment showing that a scalar reward can substitute for a demonstration, but this result is limited to a single simple task, requires online rollouts in the environment, and uses a manually specified cost function. It does not address the harder long-horizon tasks or settings where neither demonstrations nor online interaction is available. The final version should clearly outline the assumptions needed on the environment/task for their method to be practically feasible.

Finally, we note that two reviewers initially claimed the paper lacked real-robot experiments (although one later corrected their concern to lack of statistical power with n=5), but that this was verified to be incorrect.  Quantitative results appear in Table 3, the experimental setup is described in Section 5.2 and Appendix B.1, and video evidence is linked in footnote 1. These claims were not considered in reaching this decision.